# Structural insights into small-molecule agonist recognition and activation of complement receptor C3aR

Jinuk Kim[1,3,4], Saebom Ko [1,4], Chulwon Choi[1], Jungnam Bae[1], Hyeonsung Byeon [2], Chaok Seok[2] & Hee-Jung Choi [1✉]

## Abstract

The complement system plays crucial roles in innate immunity and inflammatory responses. The anaphylatoxin C3a mediates pro-inflammatory and chemotactic functions through the G protein-coupled receptor C3aR. While the active structure of the C3a-C3aR-$G_i$ complex has been determined, the inactive conformation and activation mechanism of C3aR remain elusive. Here we report the cryo-EM structure of ligand-free, G protein-free C3aR, providing insights into its inactive conformation. In addition, we determine the structures of C3aR in complex with the synthetic small-molecule agonist JR14a in two distinct conformational states: a G protein-free intermediate, and a fully active $G_i$-bound state. The structure of the active JR14a-bound C3aR reveals that JR14a engages in highly conserved interactions with C3aR, similar to the binding of the C-terminal pentapeptide of C3a, along with JR14a-specific interactions. Structural comparison of C3aR in the apo, intermediate, and fully active states provides novel insights into the conformational landscape and activation mechanism of C3aR and defines a molecular basis explaining its high basal activity. Our results may aid in the rational design of therapeutics targeting complement-related inflammatory disorders.

**Keywords** C3a Receptor; Activation Mechanism; JR14a; G Protein Signaling; cryo-EM Structure
**Subject Categories** Immunology; Membranes & Trafficking; Structural Biology

## Introduction

The complement system, a key component of innate immunity, plays a crucial role in the defense against pathogens and in clearing immune complexes and apoptotic cells (Gasque, 2004). Dysregulation of the complement system has been implicated in various inflammatory disorders, such as asthma, rheumatoid arthritis, and sepsis (Laumonnier et al, 2017). Anaphylatoxins C3a and C5a, generated during complement activation, exert their pro-inflammatory and chemotactic effects by activating their respective G protein-coupled receptors (GPCRs), C3aR and C5aR1 (Ames et al, 1996; Boulay et al, 1991). These receptors are expressed on various immune cells, including monocytes, basophils, eosinophils and mast cells (Bischoff et al, 1990; Daffern et al, 1995; Füreder et al, 1995; Martin et al, 1997; Nilsson et al, 1996).

Recent structural studies on C3aR and C5aR1 have greatly advanced our understanding of anaphylatoxin recognition by their receptors. The active structures of C5a-bound C5aR1 coupled to $G_i$ and the inactive structure of antagonist-bound C5aR1 revealed specific ligand recognition and conformational changes associated with C5aR1 activation (Feng et al, 2023; Liu et al, 2018; Wang et al, 2023; Yadav et al, 2023; Yang et al, 2024). Because C5aR1 plays a pivotal role in the late stage of the C3 complement cascade, C5aR1 has been studied as an important drug target in a wide range of immune-related diseases (Ghosh and Rana, 2023). For C3aR, active structures of $G_{i/o}$-coupled C3aR in complex with three different peptide agonists, including the endogenous ligand C3a, have been reported, providing a molecular basis for the C3a selectivity of C3aR (Wang et al, 2023; Yadav et al, 2023). Additionally, structures of the apo state of C3aR coupled to $G_{i/o}$ have demonstrated a high level of basal activity of C3aR (Wang et al, 2023; Yadav et al, 2023). However, the inactive conformation of C3aR remains unclear. Understanding the molecular basis of C3aR activation and the high basal activity of C3aR is crucial for the development of effective antagonists to treat complement-related inflammatory diseases.

SB290157 was initially developed to investigate the role of C3a in inflammatory processes and treatment of allergic asthma. It antagonizes C3aR activity in response to C3a (Ames et al, 2001). JR14a, a thiophene derivative of SB290157, has been reported to inhibit C3a-induced intracellular calcium release and mast cell degranulation, indicating its potential role as a C3aR antagonist. In a previous study, JR14a showed greater inhibitory activity than SB290157, with $IC_{50}$ values in the nanomolar range (Rowley et al, 2020). However, a comprehensive understanding of the binding mode and conformational changes associated with its binding to C3aR remains unclear.

[1]Department of Biological Sciences, Seoul National University, Seoul 08826, Republic of Korea. [2]Department of Chemistry, Seoul National University, Seoul 08826, Republic of Korea. [3]Present address: Division of Biological Science and Technology, Yonsei University, Wonju 26493, Republic of Korea. [4]These authors contributed equally: Jinuk Kim, Saebom Ko. ✉E-mail: choihj@snu.ac.kr

To elucidate the structural basis of JR14a's antagonistic activity toward C3aR, we initially aimed to determine the structure of C3aR in complex with JR14a, expecting it to represent an inactive state of C3aR. However, our cell-based assays, along with recent studies, revealed that JR14a functions as an agonist of C3aR. Of note, JR14a inhibited forskolin-induced cAMP production in a dose-dependent manner, not only in HEK293 cells but also in human monocyte THP-1 cells, which naturally express functional C3aR (Luo et al, 2025). Additionally, receptor desensitization following ligand-induced stimulation may explain its antagonist-like effects, as evidenced by the abolition of C3a-induced calcium signaling when C3a was treated 10 min after JR14a stimulation, indicating JR14a-induced receptor desensitization (Luo et al, 2025). Another study further demonstrated that SB290157 and JR14a act as a potent agonist for C3aR, with their blockade of calcium influx attributed to β-arrestin mediated internalization (Mathieu et al, 2005; Rodriguez et al, 2024). Given the absence of a known C3aR antagonist, we focused on obtaining the apo structure of C3aR in the absence of the G protein to capture its inactive state. Solving the apo structure of GPCRs remains particularly challenging owing to their intrinsic flexibility, which has resulted in only approximately 10 apo structures being determined to date, in contrast to approximately 160 unique GPCR structures in the G protein-bound active states (Munk et al, 2016).

Here, we report the cryo-electron microscopy (cryo-EM) structures of C3aR in three distinct conformational states: apo, intermediate, and fully active. We determined the structures of the ligand-free and G protein-free apo state of C3aR at 3.6 Å resolution, the JR14a-bound C3aR in the absence of G protein at 3.5 Å resolution, and the JR14a-C3aR-G protein complex at 2.5 Å resolution. During manuscript revision, the structures of JR14a-bound C3aR in both G protein-bound and G protein-free states were published. Notably, the JR14a-bound, G protein-free structure resembles the G protein-bound active state structure (Luo et al, 2025), whereas our apo and intermediate structures exhibited key characteristics of the inactive conformation. As only active C3aR structures have been reported previously, our work is essential for defining its inactive conformation. By comparing the apo, intermediate, and active structures, we propose an activation mechanism for C3aR and elucidate the structural basis of JR14a agonism. Our findings provide molecular insights into the conformational landscape of C3aR and may facilitate the rational design of novel therapeutic agents targeting C3aR.

## Results

### Apo, intermediate, and fully active structures of C3aR

To date, only the active state structures of C3aR have been reported, including the ligand-free apo state of C3aR in complex with $G_{i/o}$, which represents its active conformation (Wang et al, 2023; Yadav et al, 2023). Therefore, due to the lack of an inactive structure, the conformational changes associated with C3aR activation upon agonist binding remain poorly understood. We initially attempted to capture the inactive conformation of C3aR using JR14a, which had been identified as a C3aR antagonist (Rowley et al, 2020). However, our BRET and cAMP accumulation assays, which monitor $G_i$ recruitment to C3aR and $G_i$ signaling upon JR14a treatment, revealed a dose-dependent increase in the BRET ratio with an $EC_{50}$ of 5 nM and a decrease in cAMP levels with an $IC_{50}$ of 4 nM (Fig. EV1; Appendix

Table S1). These results indicate that JR14a acts as a C3aR agonist rather than an antagonist.

Thus, we changed our approach and determined the structure of the ligand-free and G protein-free apo state of C3aR, to capture its inactive conformation. Given that C3aR exhibits high basal activity, and forms a complex with the G protein in the absence of an agonist, we engineered a C3aR-BRIL construct, in which ICL3 was replaced with the thermostabilized apocytochrome b562 protein (BRIL) to stabilize the receptor in an inactive conformation. Then, C3aR-BRIL was purified in complex with anti-BRIL Fab and an anti-Fab nanobody. The structure of this complex was determined using cryo-EM at a local resolution of 3.6 Å in the transmembrane domain (TMD) (Figs. 1A and EV2; Appendix Table S2). To further investigate the molecular basis of C3aR activation upon JR14a binding, we determined the structures of JR14a-bound C3aR-BRIL to capture an agonist-bound intermediate state (Figs. 1B and EV3; Appendix Table S2), as well as the fully active JR14a-C3aR-$G_i$ complex (Figs. 1C and EV4; Appendix Table S2).

The structure of the JR14a-C3aR-$G_{i1}$ complex exhibits typical features of active structure of class A GPCRs, including a 5 Å outward movement of TM6 and a 5 Å inward shift of TM7 at the cytoplasmic side, compared to the apo structure (RMSD = 1.7 Å for 206 Cα atoms) (Fig. 1D). In contrast, JR14a-bound C3aR-BRIL shows high structural similarity to apo C3aR-BRIL, with an RMSD of 0.5 Å for 224 Cα atoms, adopting an inactive conformation, characterized by a closed intracellular side where TM6 remains close to TM3 (Fig. 1D). This inactive state was further supported by the conformation of conserved microswitches, such as the DRY, PIF, and NPxxY motifs, as well as the toggle switch, all of which exhibited an inactive-like configuration in both the JR14a-bound C3aR-BRIL and apo C3aR-BRIL structures (Fig. 1E). Notably, in these inactive conformations, helix 8 (H8) of C3aR (450–458) is oriented toward the cytoplasmic center of the transmembrane domain (TMD) in a reverse orientation compared with conventional H8 positioning in GPCRs. This unique H8 conformation is stabilized by a hydrophobic interaction network involving residues on H8 and TMs 1, 2, and 7. Specifically, F458^H8 interacts with F61^2.43 and Y435^7.53, whereas L454^H8 and I450^H8 form contacts with L43^1.52, V44^1.53, and V47^1.56 (Appendix Fig. S1A). This conformation closely resembles the inactive structure of C5aR1 bound to an antagonist, PMX53 (Liu et al, 2018), in which a similar hydrophobic interaction network is observed, involving L323^H8, L315^H8, and L319^H8 on H8 and L57^1.52, V58^1.53, V61^1.56, F75^2.43, and Y300^7.53 in the TMD (Appendix Fig. S1B). Sequence alignment of C3aR (449–462) with C5aR1 (314–327) reveals a strong conservation of hydrophobic residues on H8 (Appendix Fig. S1D). Mutation of F458^H8 to Ala or deletion of the F458^H8 region (1–449) greatly reduced surface expression (Appendix Fig. S1E), suggesting that F458^H8 contributes to the overall structural stability of C3aR. As GPCRs that are incompletely folded or misfolded fail to pass the ER quality control mechanism and are marked for degradation (Dong et al, 2007), a significantly low level of surface expression could indicate that GPCRs are structurally unstable.

Although the overall architecture of JR14a-C3aR-BRIL represents an inactive conformation, JR14a binding induces conformational changes in the extracellular segments of C3aR compared to the inactive apo structure of C3aR. Specifically, while a significant portion of ECL1 and ECL2 was unresolved in the apo structure, likely due to the high flexibility of these extracellular regions in the

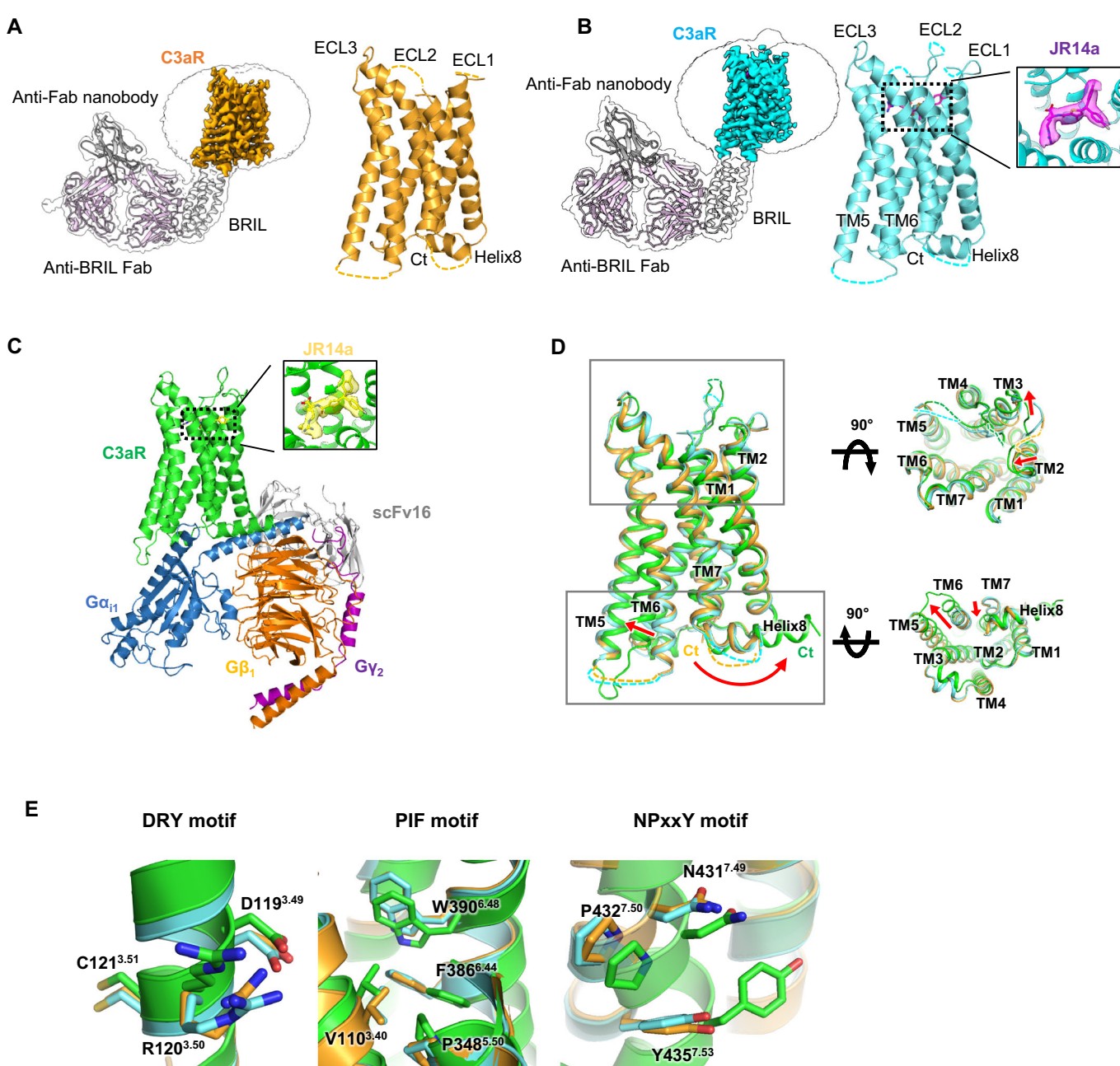

**Figure 1. Cryo-EM structures of apo, intermediate and fully active JR14a-bound C3aR.**

(A) Cryo-EM density map (left) and model (right) of apo state C3aR. The portions of BRIL, anti-BRIL Fab, and anti-Fab nanobody were displayed as cartoon. Unresolved regions in the model are displayed as dashed lines. C3aR, BRIL, anti-BRIL Fab, and anti-Fab nanobody are colored bright orange, white, light pink, and dark gray, respectively. (B) Cryo-EM density map (left) and model (right) of JR14a-bound intermediate C3aR. The portions of BRIL, anti-BRIL Fab, and anti-Fab nanobody were displayed as cartoon. Unresolved regions in the model are displayed as dashed lines. JR14a, shown as magenta sticks, with the surrounding cryo-EM density is highlighted in a zoomed-in view. The color scheme for each protein is as follows: C3aR (cyan), BRIL (light gray), anti-BRIL Fab (light pink), and anti-Fab nanobody (dark gray). (C) Model of JR14a-bound active C3aR − $G_i$ complex. The color scheme for each protein is C3aR (green), $G\alpha_{i1}$ (skyblue), $G\beta_1$ (orange), $G\gamma_2$ (purple), and scFv16 (light gray). JR14a, shown as yellow sticks, with the surrounding cryo-EM density is shown in a zoomed-in view. (D) Structural alignment of apo (bright orange), JR14a-bound intermediate (cyan) and JR14a-bound active (green) C3aR is presented in the front (left), extracellular (upper right), and intracellular (bottom right) views. Red arrows represent the movements of the TM helices and helix 8. (E) Structural changes of conserved microswitches DRY, PIF and NPxxY motifs are shown.

absence of a ligand, a specific region of ECL2 was well-resolved in the JR14a-C3aR-BRIL structure, by participating in the interaction with JR14a. However, these conformational changes did not propagate into the TMD, including the conserved microswitch region, resulting in an overall inactive conformation (Fig. 1E). Thus, our structural analysis suggests that JR14a-bound C3aR-BRIL represents an early intermediate state of C3aR, an agonist-bound C3aR that has not yet undergone full activation by G protein

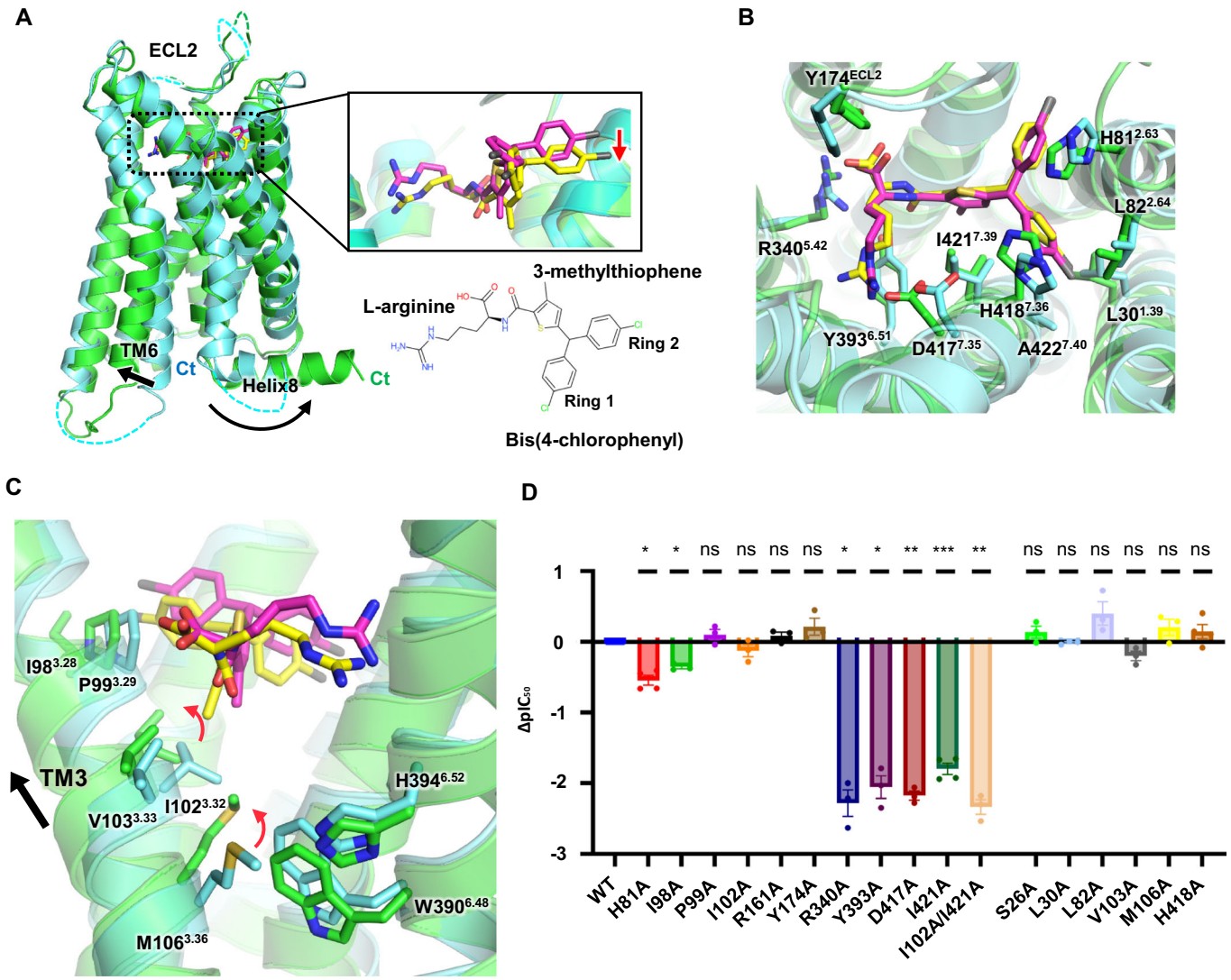

**Figure 2. Comparison of JR14a bound intermediate and active states.**

(A) An overlay of JR14a-bound intermediate structure (cyan) and the active structure (green) of C3aR is shown. Movements of TM6 helix and Helix8 are indicated by black arrows. Unresolved regions in the intermediate structure are displayed as dashed lines. Close-up view of JR14a in intermediate (magenta) and active (yellow) structures is shown. The movement of JR14a is indicated by red arrow. The chemical structure of JR14a with brief description of moieties is shown below. (B) Key interactions within the JR14a binding pocket are shown. Residues interacting with JR14a in both intermediate and active structures are represented as sticks. (C) Detailed interactions between C3aR TM3 and JR14a in two structures are shown. The movement of TM3 is indicated by black arrow, while the distinct movements of I102$^{3.32}$ and M106$^{3.36}$ in active structure are indicated by red arrow. (D) Mutagenesis study of C3aR for cAMP response by JR14a treatment. For each C3aR mutant, $\Delta pIC_{50}$ was calculated against WT C3aR. Error bars represent the standard error of the mean (S.E.M.) from $n = 3$–4 independent experiments, with detailed $n$ values provided in Appendix Table S1. Statistical analysis was performed using one-way ANOVA with Dunnett's multiple comparisons test compared to WT. The $P$ values are denoted as ns ($P > 0.05$), * ($P \leq 0.05$), ** ($P \leq 0.01$), *** ($P \leq 0.001$). (The $P$ values of each mutant vs. WT are; H81A, $P = 0.0145$; I98A, $P = 0.0161$; P99A, $P = 0.7135$; I102A, $P = 0.6851$; R161A, $P = 0.6125$; Y174A, $P = 0.5373$; R340A, $P = 0.0182$; Y393A, $P = 0.0171$; D417A, $P = 0.0024$; I421A, $P = 0.0007$; I102A/I421A, $P = 0.0055$; S26A, $P = 0.6073$; L30A, $P = 0.9323$; L82A, $P = 0.3611$; V103A, $P = 0.266$; M106A, $P = 0.5401$; H418A, $P = 0.698$). Source data are available online for this figure.

coupling. This is similar to the structure of the agonist-bound β₂-adrenergic receptor (β₂AR) in the absence of G protein (Rosenbaum et al, 2011).

## Comparison of JR14a binding pockets in the intermediate and active states

The cryo-EM map showed clear JR14a density in the ligand-binding pocket of both the intermediate and active structures of

C3aR (Figs. 1B,C, EV3 and EV4). Notably, although the binding mode of JR14a remains largely conserved between the two states, JR14a is positioned ~1.5 Å deeper in the fully active structure than in the intermediate state (Fig. 2A).

In both the structures, JR14a maintains its key interactions with C3aR. The arginine moiety and carboxyl group of JR14a, which mimic the C-terminal Arg of C3a, form ionic interactions with D417$^{7.35}$ and R340$^{5.42}$, respectively. Additionally, JR14a interacts with several ring-structured residues H81$^{2.63}$, Y174$^{ECL2}$, Y393$^{6.51}$,

and H418$^{7.36}$, and hydrophobic residues L30$^{1.39}$, L82$^{2.64}$, I421$^{7.39}$, and A422$^{7.40}$ of C3aR via its 3-methylthiophene and two 4-chlorophenyl groups (Fig. 2B). However, in the fully active structure, JR14a forms additional contacts because of its deeper binding, resulting in more extensive hydrophobic interactions—particularly with TM3 residues, such as I98$^{3.28}$, P99$^{3.29}$, I102$^{3.32}$, V103$^{3.33}$, and M106$^{3.36}$. This is facilitated by the deeper binding of JR14a and the resulting TM3 movement. Specifically, in the active structure, I102$^{3.32}$ interacts with both the 3-methylthiophene ring and one of the 4-chlorophenyl rings of JR14a, whereas in the JR14a-C3aR-BRIL structure, I102$^{3.32}$ interacts only with the 3-methylthiophene ring (Fig. 2C). Notably, M106$^{3.36}$ exhibits distinct rotamer configurations in the two structures. In the G$_i$-coupled active structure, M106$^{3.36}$ adopts an "up" configuration, directly interacting with the methyl group of JR14a. In contrast, in the BRIL-fused intermediate structure, M106$^{3.36}$ has a "side" configuration, oriented toward TM5 and TM6, where it interacts with W390$^{6.48}$ and H394$^{6.52}$ instead of JR14a (Fig. 2C). To assess whether these conformational differences affect the ligand-binding pocket volume, we calculated the pocket volume using a previously reported method for β$_1$AR ligand binding studies (Warne et al, 2019). Our results indicate that the JR14a-bound intermediate state has a larger ligand-binding pocket (647 Å$^3$) than the G protein-bound active structure (575 Å$^3$). This decrease in the volume of the ligand-binding site in the active state is likely driven by an allosteric effect from G protein coupling, which induces conformational changes such as the upward rotamer shift of M106$^{3.36}$ (Appendix Fig. S2). These findings align with β$_1$AR study, which showed a correlation between a decrease in the volume of the orthosteric binding pocket and an increase in agonist-binding affinity in the active state (Warne et al, 2019).

Mutagenesis studies support our structural findings, demonstrating that the interactions mediated by H81$^{2.63}$, I98$^{3.28}$, R340$^{5.42}$, Y393$^{6.51}$, D417$^{7.35}$, and I421$^{7.39}$ of C3aR are crucial for G$_i$ signaling upon JR14a binding. Substituting these residues with Ala significantly reduced the downstream responses (Figs. 2D and EV5; Appendix Table S1). Single point mutations of other residues (P99$^{3.29}$, R161$^{ECL2}$ and Y174$^{ECL2}$) into Ala displayed relatively mild effects on downstream signaling, with less than a 10-fold decrease in IC$_{50}$ values upon JR14a binding.

## Comparison of binding modes of JR14a, EP54, and C3a

Three agonist-bound active structures of C3aR coupled with G$_{i/o}$ have been reported previously. These include C3aR bound to the natural peptide ligand C3a (a 77-amino acid peptide), the 15-amino acid peptide EP141, and the 10-amino acid peptide EP54 (Wang et al, 2023; Yadav et al, 2023). However, structure of C3aR bound to a small-molecule agonist has not been reported until recently. Given that the small-molecule agonist JR14a exhibits signaling activity comparable to that of the decapeptide agonist EP54 (Fig. EV1), a structural analysis comparing the binding of JR14a and peptide agonists would be beneficial for designing small molecule C3aR agonists or antagonists.

The C-terminal Arg residues of the peptide agonists (R77 of C3a and R10 of EP54) are critical for C3aR activation (Bajic et al, 2013; Wilken et al, 1999). Similarly, JR14a contains an arginine moiety with a carboxyl group, forming polar contacts with C3aR residues, such as Y174$^{ECL2}$, R340$^{5.42}$, Y393$^{6.51}$, and D417$^{7.35}$, which were also

observed in the peptide agonist-bound structures (Fig. 3A,B; Appendix Fig. S3A). Mutagenesis studies have shown that mutations R340$^{5.42}$A, Y393$^{6.51}$A, and D417$^{7.35}$A significantly reduce downstream G$_i$ signaling in response to C3a, with a similar reduction observed for JR14a (Appendix Fig. S4 and Appendix Table S1). However, the R161$^{ECL2}$A and Y174$^{ECL2}$A mutants responded differently: signaling was greatly reduced for C3a but not for JR14a (Appendix Fig. S4 and Appendix Table S1). This can be attributed to the stable contacts between R161$^{ECL2}$ and Y174$^{ECL2}$ of C3aR and the C-terminal carboxyl group as well as the L75 carbonyl group and L73 of C3a, which correctly position these residues for interaction with C3a. In contrast, in the JR14a-bound structure, R161$^{ECL2}$ and Y174$^{ECL2}$ lack these additional stabilizing interactions with JR14a, making their contacts more transient due to ECL2's inherent flexibility (Appendix Fig. S5).

The 3-methylthiophene moiety of JR14a occupies a position analogous to the side chains of A76 in C3a and D-Ala in EP54. However, it establishes a more extensive hydrophobic interaction network with I102$^{3.32}$, V103$^{3.33}$, M106$^{3.36}$, and I421$^{7.39}$ (Fig. 3A,B; Appendix Fig. S3B). Notably, while I421$^{7.39}$ forms van der Waals contact with A76 of C3a, it engages in more extensive interactions with the 4-chlorophenyl ring (Ring 1) and the 3-methylthiophene moiety of JR14a. In line with this, the I421$^{7.39}$A mutant exhibited a minor effect on C3a-mediated signaling but caused a 60-fold reduction in JR14a-induced signaling, underscoring the important role of I421$^{7.39}$ in JR14a binding (Figs. 2D and EV5; Appendix Fig. S4 and Appendix Table S1).

One of the two 4-chlorophenyl rings (Ring 2) in JR14a aligns with L75 of C3a and L9 of EP54, forming similar interactions with H81$^{2.63}$, P99$^{3.29}$, and I102$^{3.32}$. However, the bulkier 4-chlorophenyl moiety of JR14a forms additional close contact with I98$^{3.28}$. Notably, W88$^{ECL1}$ on ECL1, which forms van der Waals contact with L75 of C3a and L9 of EP54 in their respective structures, is pushed away by the chloride group of JR14a yet still maintains contact with JR14a. This indicates a degree of structural plasticity in ECL1 for accommodating different ligands (Fig. 3A,B; Appendix Fig. S3C).

The other 4-chlorophenyl ring in JR14a (Ring 1) does not overlap with any segments of C3a and EP54, interacting with S26$^{1.35}$, L30$^{1.39}$, and L82$^{2.64}$, which are not involved in interactions with C3a or EP54 (Fig. 3A,B; Appendix Fig. S3C). However, single point mutations of these residues to Ala did not cause significant reduction in downstream G$_i$ signaling (Fig. 2D; Appendix Table S1), likely due to the extensive hydrophobic and polar interactions of the remaining groups of JR14a.

Altogether, our structural analysis highlights how JR14a achieves comparably high activity to C3a, despite its smaller size and fewer interactions with ECLs. Unique interactions with the 4-chlorophenyl rings of JR14a and enhanced interactions with I421$^{7.39}$ of C3aR likely contribute to JR14a's potency compared to the C3a 15-amino acid peptide (63–77). These findings could provide valuable insights for the design of high-affinity small-molecule agonists targeting C3aR.

## Molecular basis for basal activity of C3aR

Recent studies have revealed that C3aR exhibits high basal activity, indicating that it can be activated without agonist binding, as evidenced by the structures of ligand-free C3aR coupled to G$_{i/o}$

**A**

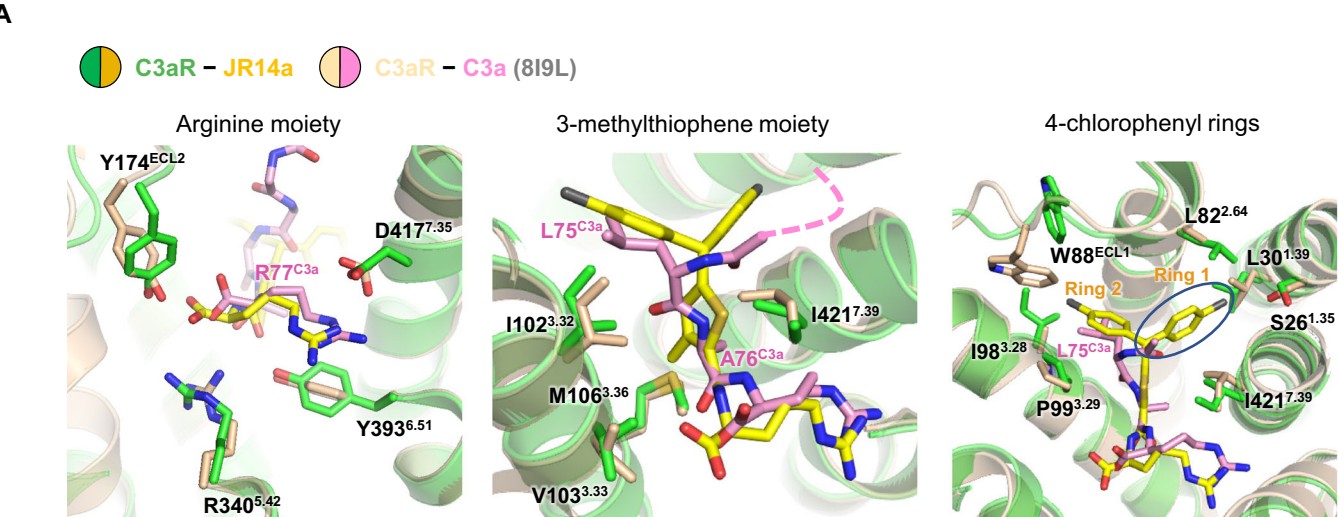

**B**

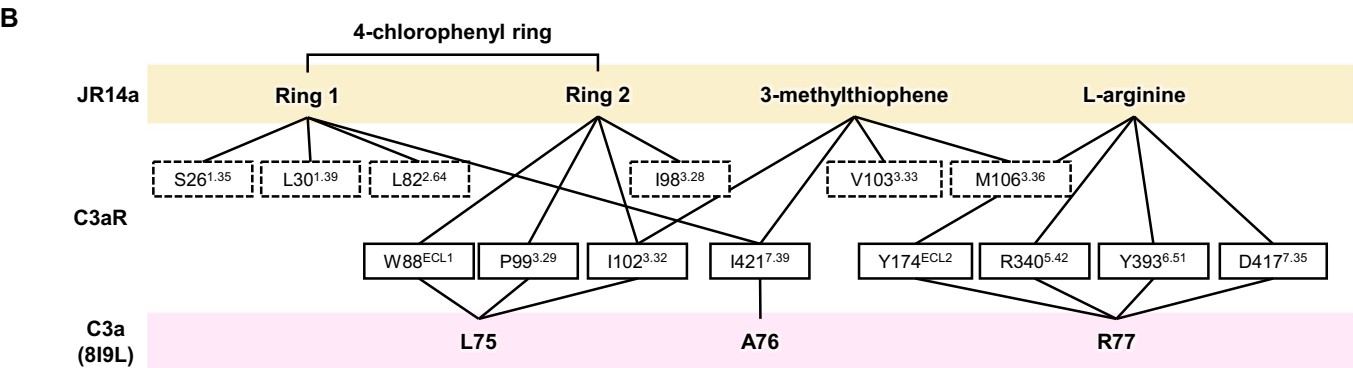

**Figure 3.   Comparison of the binding modes of JR14a, EP54, and C3a.**

(A) Structural superposition of JR14a-bound active state C3aR with C3a-bound C3aR (PDB: 8I9L), focusing on interactions involving the C-terminal arginine moiety of JR14a (left). JR14a, C3a, and interaction residues are labeled in the figure with corresponding colors, as indicated above. Interactions involving the 3-methylthiophene (middle) and 4-chlorophenyl rings (right, denoted as Ring 1 and Ring 2 in the figure) moieties of JR14a are shown. JR14a unique binding site is indicated by blue ellipse. For visual clarity, part of C3a is replaced with dashed line or omitted. (B) Diagram comparing the interactions of the JR14a and native ligand C3a within the orthosteric binding pocket of C3aR. Shared interacting residues are shown in solid boxes, while JR14a-specific interactions are highlighted with dashed boxes. Ligand colors correspond to those indicated above.

(Wang et al, 2023; Yadav et al, 2023). In general, apo state GPCR samples a diverse conformational landscape, fluctuating between inactive and active-like states (Elgeti and Hubbell, 2021), with agonist binding shifting the conformational equilibrium towards the active state. In the case of C3aR, the energy barrier for transitioning to the active state appears to be lower, contributing to its high basal activity. To elucidate the structural basis of C3aR's constitutive activity, we compared our inactive apo C3aR structure with that of C5aR1, a closely related class A GPCR with minimal basal activity, focusing on the key structural elements that may contribute to the constitutive activity of C3aR.

Our analysis of the apo C3aR structure revealed several features likely contributing to its basal activity. The ionic lock formed between $R^{3.50}$ and $D/E^{6.30}$, a structural feature observed in some class A GPCRs, has been proposed to stabilize the inactive state (Dore et al, 2011; Shao et al, 2016; Wang et al, 2018; Zhou et al, 2019). However, in C3aR, at these positions, $R120^{3.50}$ and $Q372^{6.30}$ are

present, which do not interact with each other, as observed in our apo structure (Fig. 4A). This differs from C5aR1, where $S237^{6.30}$, although present instead of $D/E^{6.30}$, still interacts with $R134^{3.50}$, stabilizing the inactive conformations of TM3 and TM6 (Fig. 4A). The absence of an ionic lock (or a similar stabilizing interaction) in C3aR may lower the energy barrier for TM6 to shift into an open conformation, even in the absence of an agonist, facilitating G protein binding and subsequent activation.

Another structural feature related to C3aR's basal activity is the instability of the hydrophobic network beneath the ligand-binding pocket. In C5aR1, this network is composed of tightly packed hydrophobic residues, such as $I116^{3.32}$, $V286^{7.39}$, $L89^{2.57}$, $F44^{1.39}$, $L92^{2.60}$, and $Y290^{7.43}$. In C3aR, however, the highly hydrophobic $F44^{1.39}$ of C5aR1 is replaced by the less hydrophobic $L30^{1.39}$, whereas $L92^{2.60}$ and $Y290^{7.43}$ are replaced with the polar residues $S78^{2.60}$ and $S425^{7.43}$ (Fig. 4B). Notably, sequence analysis of class A GPCRs bound to peptide ligands demonstrates that large hydrophobic

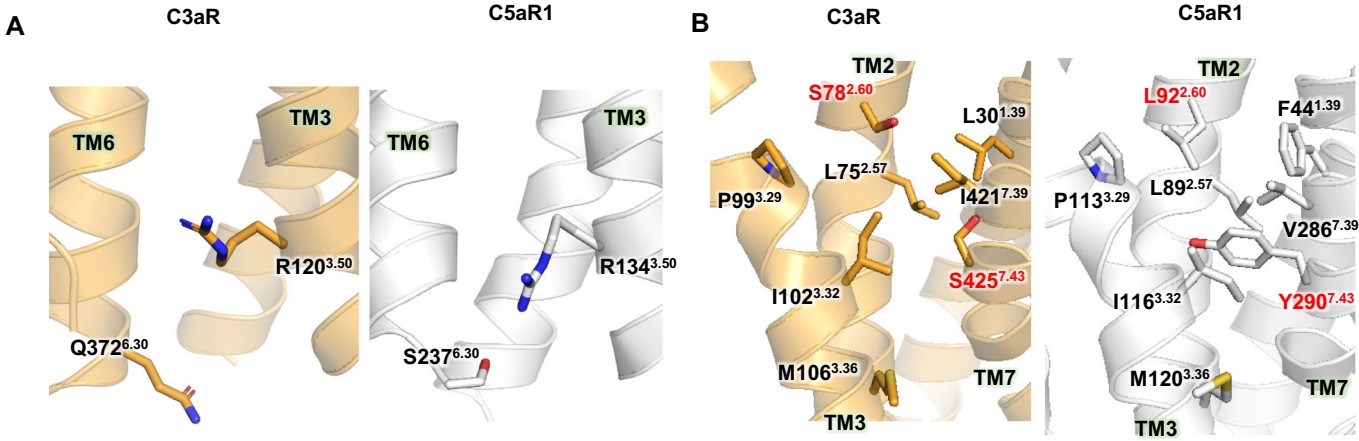

**Figure 4. Structural analysis of the apo state of C3aR compared to inactive C5aR1.**

(A) Comparison of the position of R3.50 and Q6.30 in apo C3aR (bright orange) and R3.50 and S6.30 in inactive C5aR1 (white, PDB: 6C1R). Residues at these positions are shown as sticks. (B) Loose hydrophobic packing underneath the ligand binding pocket of C3aR compared to C5aR1. Residues S78^2.60 and S425^7.43 in C3aR and their counterparts in C5aR1 are highlighted in red.

residues, such as Phe or Tyr are commonly found at the 7.43 position (Appendix Fig. S6). The presence of polar residues at these positions in C3aR results in a looser hydrophobic packing, which reduces the structural stability of the inactive state and enhances the structural dynamics of the receptor, thereby increasing the likelihood of adopting active-like states, and contributing to its high basal activity. Notably, this structural characteristic of C3aR bears similarity to the M120^3.36A/I116^3.32A mutant of C5aR1, which exhibits basal activity (Feng et al, 2023). In this C5aR1 mutant, the replacement of bulky and hydrophobic residues with alanine led to a less constrained hydrophobic core, weakening the structural forces that stabilize the inactive state.

Based on all these results, we propose that sequence variations and consequent structural features within the TMD of C3aR may account for its high basal activity. However, the relationship between these structural features and basal activity cannot be generalized across all GPCRs. For example, while the E410^6.30K mutation in the histamine H1 receptor (H1R), which possesses an ionic lock, increases basal activity (Ma et al, 2021), introducing the A298^6.30E mutation in H4R, which lacks an ionic lock, does not reduce basal activity (Schneider et al, 2010). These results highlight the complex interplay of interactions within the TMD and receptor-specific structural features that contribute to GPCR activation.

## Activation mechanism of C3aR upon JR14a binding

The activation of C3aR by JR14a involves a series of conformational changes, transitioning from its apo state to the fully active state. Structural analysis of the three states of C3aR in our study illustrates how JR14a binding induces conformational changes, leading to C3aR activation.

Comparison of JR14a-bound C3aR (without G_i) to our apo C3aR structure reveals localized conformational changes near the orthosteric binding site. A notable change is the disruption of the I102^3.32–I421^7.39 interaction, caused by the insertion of the

3-methylthiophene moiety of JR14a between these residues (Fig. 5A). The I102^3.32A/I421^7.39A double mutation synergistically reduced JR14a-induced signaling, highlighting the importance of these two residues. Although individual I102^3.32A and I421^7.39A mutations had little effect on C3a-induced signaling, this double mutation led to a decrease, suggesting a cooperative role of I102^3.32 and I421^7.39 in receptor activation. Additionally, highly unstructured ECL2 in the apo state is partially resolved in the JR14a-bound state, particularly around residues R161^ECL2 and Y174^ECL2, which interact directly with JR14a. Similarly, W88^ECL1 is modeled in the JR14a-bound structure due to its interaction with JR14a (Fig. 5A). JR14a also interacts with residues on TM3, such as I98^3.28, P99^3.29, and I102^3.32, through its distinct molecular groups, the 4-chlorophenyl ring and the 3-methylthiophene group, inducing a slight rotational movement of the extracellular portion of TM3 (Fig. 5A).

Deeper binding of JR14a into the TMD induces substantial conformational shifts in TM3, TM6, and TM7, facilitating G protein coupling and driving the transition to a fully active state. Specifically, to avoid steric clashes between I102^3.32 and the methyl group of JR14a's 3-methylthiophene moiety, TM3 undergoes further rotational movement, along with a translational shift. This causes M106^3.36 to adopt an "up" rotamer configuration, allowing interaction with the methyl group of JR14a. In the inactive state, M106^3.36 interacts with W390^6.48 in a "side" configuration (Fig. 5B). Upon activation, this rotamer change triggers repacking of the surrounding side chains, including the downward movement of the W390^6.48 toggle switch, which further propagates to rearrangement of the cytoplasmic region of TM6, a key step in the activation process. During this process, the significant inward movement of Y435^7.53 disrupts the aromatic interaction network with F61^2.43 and F458^H8 (Fig. 5C), which stabilized the inactive state, leading to the release of H8. In the active state, F442^H8 replaces L454^H8, which interacts with V44^1.53 in the inactive state, and forms a new hydrophobic interaction network with V44^1.53, A48^1.57, and L437^7.55 (Fig. 5C). As H8 is released from the intracellular center of

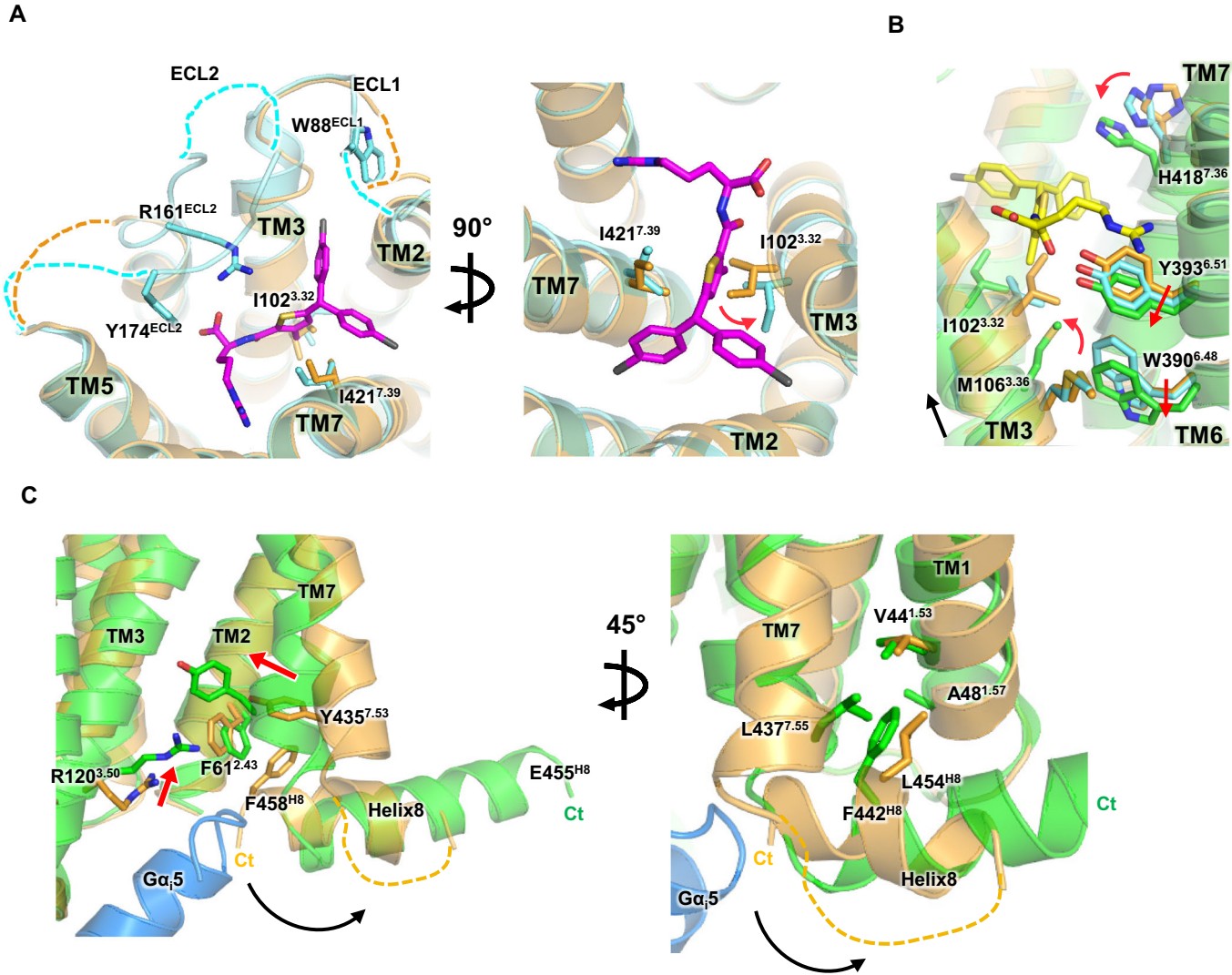

**Figure 5. Activation mechanism of C3aR upon JR14a binding.**

(A) Structural comparison of apo C3aR (bright orange) and JR14a-bound intermediate C3aR (cyan). JR14a (magenta) binding induces ordering in parts of ECL1 and ECL2, with unstructured regions shown as dashed lines (left). The movement of I102$^{3.32}$ is highlighted by red arrows in the rotated view (right). (B) Structural comparison of apo (bright orange), intermediate (cyan), and active (green) C3aR, with JR14a in the active structure shown in yellow. The movement of TM3 is indicated by black arrow, while the movements of M106$^{3.36}$, W390$^{6.48}$, Y393$^{6.51}$, and H418$^{7.36}$ are highlighted by red arrows. (C) Structural comparison of the intracellular G protein binding site of apo and active state C3aR, with the Gα$_i$5 helix shown in skyblue. The movements of R120$^{3.50}$ and Y435$^{7.53}$ are indicated by red arrows, while the movement of helix8 is shown by black arrow. Residues near helix8 involved in interactions are depicted as sticks.

the TMD, the intracellular cavity opens, allowing the Gα5 helix of the G protein to insert – a characteristic feature of activation in most class A GPCRs.

A comparison between the apo-C3aR-G$_o$ and JR14a-C3aR-G$_i$ complexes reveals that the two TMDs are highly similar, particularly around TM3, near I102$^{3.32}$ and M106$^{3.36}$. Notably, M106$^{3.36}$ adopts an upright configuration in all active C3aR structures, including in the apo-C3aR-G$_o$ complex. The differences between these two structures lie primarily in the side chain positions of the residues in the ECLs and the ligand-binding pocket, where JR14a interacts directly. For instance, the rotamers of R340$^{5.42}$ and Y393$^{6.51}$ are rearranged to interact with JR14a properly, and W88$^{ECL1}$ and R161$^{ECL2}$ on ECLs are repositioned for optimal

JR14a binding in the JR14a-bound fully active state (Appendix Fig. S7).

Altogether, the structural analysis of the three C3aR states suggests a potential sequence of conformational changes triggered by agonist binding. Upon agonist interaction, localized structural changes occur in the ECLs and ligand-binding pocket, leading to TMD rearrangement and subsequent G protein coupling (Fig. 6).

## Discussion

In this study, we present three distinct structural states of C3aR: the apo state, representing the inactive conformation; the JR14a-bound

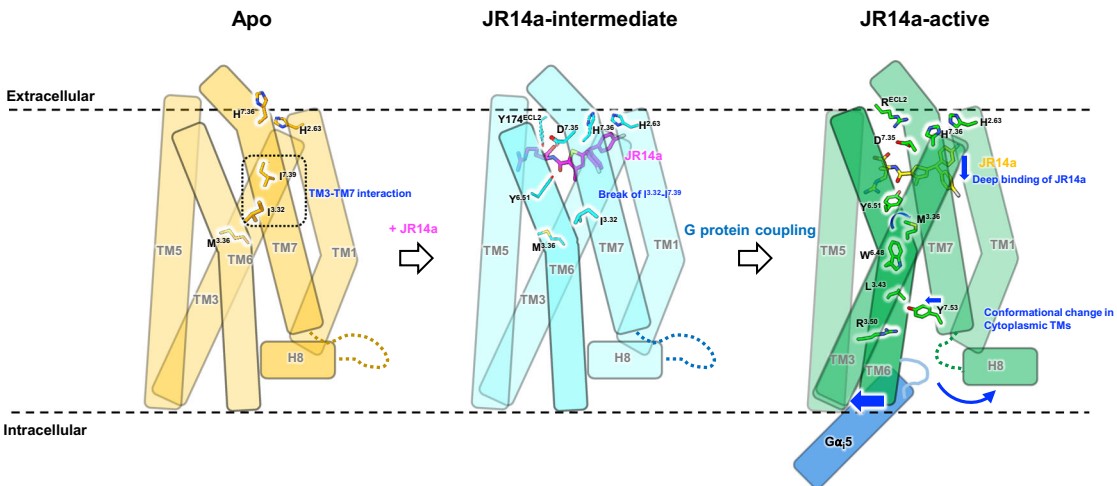

**Figure 6.  Schematic model of C3aR activation mechanism upon JR14a binding.**

The transition from the apo (bright orange), to the JR14a-intermediate (cyan) and fully active (green) states of C3aR is illustrated. In the apo state, an interaction between I102$^{3.32}$ and I421$^{7.39}$ is present. JR14a binding disrupts this interaction, causing structural changes in ECL2 and the orthosteric binding pocket, leading to an intermediate state. Further deeper binding of JR14a along with G protein coupling drives full activation. JR14a is shown in magenta (intermediate) and yellow (active), with major movements indicated by blue arrows. Key residues are depicted as sticks.

state in the absence of G protein, capturing an intermediate conformation; and the JR14a-C3aR-G$_i$ complex, reflecting the fully active conformation. Our apo C3aR structure represents the first inactive state of this receptor, serving as a valuable reference for understanding the conformational changes associated with C3aR activation. Additionally, the structure of JR14a-bound, G protein-free C3aR displays an inactive conformation within the TMD but an active-like conformation in the ECLs. This suggests that this structure represents an early-stage intermediate conformation following JR14a binding, providing insights into the initial conformational changes induced by agonist binding. In both the apo and intermediate states, H8 adopts a reversed orientation toward the cytoplasmic center of the TMD, a feature previously observed in antagonist-bound inactive structures of C5aR1. In the fully active structure of the JR14a-C3aR-G$_i$ complex, JR14a binds deeper into the ligand-binding pocket compared to the intermediate state. Recently, the structures of JR14a-bound C3aR-BRIL and JR14a-C3aR-G protein complex have been reported, but both were classified as active conformations (Luo et al, 2025).

Comparison of the three structural states, together with the mutagenesis study, provides a comprehensive understanding of the activation mechanism of C3aR upon JR14a binding. Initially, JR14a binding induces local conformational changes in ECL2 and rotamer changes of residues that directly interact with JR14a within the ligand-binding pocket. As JR14a binds more deeply, as observed in the fully active structure, it induces large-scale structural rearrangements, including translational and rotational movements of TM3, repacking of residues beneath the ligand-binding pocket, and significant outward shift of TM6 along with inward movement of TM7. These changes lead to the release of H8 and the opening of the intracellular cavity for G protein coupling. This activation pathway closely resembles that of C5aR1. However, unlike C5aR1, C3aR exhibits constitutive activity, likely due to the lower stability of its inactive structure, facilitating an equilibrium shift toward activation.

The small-molecule agonist JR14a differs from C3a in size and interaction networks with C3aR. While JR14a relies on specific interactions with residues within the orthosteric pocket, such as I421$^{7.39}$, C3a forms extensive contacts across both extracellular and transmembrane regions, likely stabilizing the active conformation through a broader network of interactions. These differences suggest that the structural contributions of individual residues to receptor activation vary depending on the ligand, making it difficult to define a single key residue responsible for C3aR activation. In this context, ECL2 may play a more critical role in C3a binding than in JR14a binding, suggesting its involvement in the initial stages of receptor activation. These differences make it challenging to establish a general activation mechanism for C3aR. To fully elucidate its activation process, it is necessary to systematically analyze the structural characteristics of various intermediate states for multiple peptide ligands, including C3a. Additionally, beyond the conformational states associated with G protein coupling, it is important to examine the conformational subsets linked to β-arrestin coupling, as previously studied for C5aR1 (Gupta et al, 2024).

Our findings provide valuable insights for structure-based drug design targeting C3aR. The binding site and the specific interactions identified in the JR14a-bound structures offer crucial information for developing more potent and selective small-molecule C3aR agonists or antagonists. Furthermore, our apo structure highlights the inactive conformation of C3aR, offering a foundation for designing antagonists that stabilize this inactive conformation. The intermediate conformation captured in the JR14a-C3aR-BRIL structure demonstrates early conformational changes induced by agonist binding and provides insights into the transitions required for receptor activation. By integrating insights from the apo, intermediate, and fully active states of C3aR, these structural data establish a framework for optimizing agonists or antagonists to achieve the desired therapeutic outcomes.

# Methods

### Reagents and tools table

| Reagent/resource | Reference or source | Identifier or catalog number |
|---|---|---|
| **Experimental models** | | |
| *E. coli* ROSETTA (DE3) | Novagen | Cat# 70954 |
| *Spodoptera frugiperda* (Sf9) | Expression systems | Cat#94-001F |
| High Five | Expression systems | Cat#94-002F |
| HEK293T | ATCC | Cat#CRL-3216 |
| **Recombinant DNA** | | |
| pET-21d BRIL-Fab | This study | N/A |
| pcDNA3.1 C3aR | This study | N/A |
| pcDNA3.1 G$\alpha_{i1}$ | This study | N/A |
| pcDNA3.1 G$\beta_1$ | This study | N/A |
| pcDNA3.1 G$\gamma_2$ | This study | N/A |
| pFastBac C3aR | This study | N/A |
| pFastBac C3aR-BRIL | This study | N/A |
| pFastBac-scFv16 | This study | N/A |
| pFastBac G$\alpha_{i1}$ | This study | N/A |
| pFastBac Dual G$\beta_1$, and G$\gamma_2$ | This study | N/A |
| pET-21d anti-Fab Nanobody | This study | N/A |
| GloSensor plasmid | Promega | Cat#E2301 |
| **Antibodies** | | |
| Rabbit anti-FLAG antibody | Cell Signaling Technology | Cat#14793 |
| Anti-rabbit HRP-conjugated antibody | Enzo Life Sciences | ADI-SAB-300 |
| **Oligonucleotides and other sequence-based reagents** | | |
| Peptides | Cusabio | Custom order |
| **Chemicals, enzymes and other reagents** | | |
| JR14a | MedChemExpress | Cat#2411440-41-8 |
| C3a | Sigma-Aldrich | Cat#204881-50UG |
| ESF921 incest cell culture medium | Expression system | Cat#96-001-20 |
| Cellfectin™ II Reagent | Gibco | Cat#10352100 |
| Lauryl Maltose Neopentyl Glycol | Anatrace | Cat#NG310 |
| Cholesteryl hemisuccinate | Sigma-Aldrich | Cat#C6512 |
| n-Dodecyl-β-D-Maltopyranoside | Anatrace | Cat#D310A |
| Leupeptin | Goldbio | Cat#L-010-5 |
| Benzamidine | Sigma-Aldrich | Cat#B6506 |
| Phenylmethylsulfonyl fluoride | Sigma-Aldrich | Cat#11359061001 |
| Tris(2-carboxyethyl) phosphine hydrochloride | Goldbio | Cat#TCEP1 |
| Apyrase | NEB | Cat#M0398L |
| Dulbecco's Modified Eagle Medium (DMEM) | Cytiva | Cat#SH30243.01 |
| Fetal Bovine Serum | GW vitek | Cat#US-FBS-500 |

| Reagent/resource | Reference or source | Identifier or catalog number |
|---|---|---|
| Antibiotic-Antimycotic | Gibco | Cat#15240–062 |
| Lipofectamine 2000 | Invitrogen | Cat#11668019 |
| 4% Paraformaldehyde | Tech&Innovation | Cat#BPP-9004 |
| BSA | Bovogen Biologicals | Cat#BSAS 0.1 |
| 1-Step™ TMB-Blotting Substrate Solution | Thermofisher Scientific | Cat#34018 |
| Janus Green B | Tokyo Chemical Industry | Cat#J0002 |
| Forskolin | Sigma-aldrich | Cat#F6886 |
| CO2-independent medium | Gibco | Cat# 15420604 |
| D-Luciferin | NanoLight | Cat#306 |
| Coelenterazine h | NanoLight | Cat#301 |
| Accutase solution | Sigma-aldrich | Cat# A6964 |
| **Software** | | |
| CryoSPARC | https://cryosparc.com/ | Version 4.5.1 |
| RELION | https://relion.readthedocs.io/en/release-4.0/ | Version 4.0 |
| PHENIX | https://phenix-online.org/ | Version 1.20.1-4487 |
| COOT | www.2.mrc-lmb.cam.ac.uk/personal/pemsley/coot/ | Version 0.9.8.1 |
| UCSF Chimera | https://www.cgl.ucsf.edu/chimera/ | Version 1.17.3 |
| UCSF ChimeraX | https://www.cgl.ucsf.edu/chimerax/ | Version 1.6.1 |
| POVME2 | https://durrantlab.pitt.edu/povme2/ | Version 2.2.2 |
| GraphPad Prism | https://www.graphpad.com/scientific-software/prism/ | Version 10.1.2 |
| AlphaFold2 | https://colab.research.google.com/github/sokrypton/ColabFold/blob/main/AlphaFold2.ipynb#scrollTo=kOblAo-xetgx | ColabFold v1.5.5 |
| **Other** | | |
| FlexStation 3 multi-mode microplate reader | Molecular Devices | |
| Tristar 2 LB 942 multimode reader | Berthold | |

## Expression and purification of C3aR

Full-length human C3aR (residues 1–482) was cloned with an N-terminal HA signal sequence followed by a FLAG tag, and a C-terminal eGFP-His$_6$ tag with a HRV 3C protease cleavage site. This C3aR construct was expressed in *Spodoptera frugiperda* (Sf9) insect cells using the Bac-to-Bac baculovirus expression system (Invitrogen). Cells were harvested 60 h post-infection and lysed by dounce homogenization in lysis buffer (20 mM HEPES pH 8.0, 150 mM NaCl, and 1 mM EDTA) supplemented with protease inhibitors (1 mM phenylmethylsulfonyl fluoride (PMSF), 175 μg mL$^{-1}$ benzamidine, and 10 μM leupeptin). After centrifugation, the membrane fraction was solubilized in buffer containing 1% (w/v) lauryl maltose neopentyl glycol (LMNG, Anatrace) and 0.1% (w/v) cholesteryl hemisuccinate (CHS, Sigma-Aldrich) for 2 h at 4 °C with gentle stirring. After centrifugation, the clarified supernatant was applied to Ni-NTA resin. The column was washed with buffer containing 20 mM HEPES pH 8.0, 150 mM NaCl, 20 mM imidazole,

0.005% (w/v) LMNG, and 0.0005% (w/v) CHS. The protein was eluted with 300 mM imidazole and immediately loaded onto anti-FLAG M1 affinity resin in the presence of 2 mM CaCl$_2$. After washing with the same LMNG-containing buffer (20 mM HEPES pH 8.0, 150 mM NaCl, 20 mM imidazole, 0.005% (w/v) LMNG, and 0.0005% (w/v) CHS, 2 mM CaCl$_2$), the C3aR was eluted with buffer (20 mM HEPES pH 8.0, 150 mM NaCl, 0.005% (w/v) LMNG, and 0.0005% (w/v) CHS, 0.1 mg mL$^{-1}$ FLAG peptide, and 4 mM EDTA). The purified C3aR was concentrated to 1 mg mL$^{-1}$ in preparation for complex formation with the G$_{i1}$ heterotrimer.

## Expression and purification of C3aR-BRIL

A C3aR-BRIL fusion construct was designed using AlphaFold2 (Jumper et al, 2021) with reference to adenosine A2A receptor (A2AR) structural study (Zhang et al, 2022). The construct consisted of an N-terminal FLAG tag following the HA signal sequence, the C3aR sequence with ICL3 (residues 360–370) replaced by the BRIL sequence, and a C-terminal HRV protease cleavage site followed by an eGFP-His$_6$ tag. This C3aR-BRIL construct was expressed and purified using a protocol similar to that used for the expression and purification of the wild-type C3aR. Briefly, the construct was expressed in Sf9 cells, harvested, and lysed as described above. The membrane fraction was solubilized and purified using Ni-NTA resin. The purified C3aR-BRIL fusion protein was then treated with HRV 3C protease to remove the eGFP-His$_6$ tag in preparation for complex formation with anti-BRIL Fab and anti-BRIL Fab nanobody.

## Expression and purification of heterotrimeric G$_{i1}$

The heterotrimeric G$_i$ protein, consisting of human G$\alpha_{i1}$, His$_6$-G$\beta_1$, and G$\gamma_2$ subunits, was co-expressed in *Trichoplusia ni* (Hi5) insect cells. Cells were collected 60 h post-infection and lysed in lysis buffer containing 20 mM Tris-HCl pH 8.0 and protease inhibitors (1 mM PMSF, 175 µg mL$^{-1}$ benzamidine, and 10 µM leupeptin). Following centrifugation at 18,000 rpm for 15 min at 4 °C, the G$_i$ protein heterotrimer was extracted from the membrane fraction using a solubilization buffer composed of 20 mM Tris-HCl pH 8.0, 100 mM NaCl, 2 mM MgCl$_2$, 50 µM GDP, 1% (w/v) sodium cholate, 0.1 mM TCEP, and protease inhibitors. The solubilized protein was purified by Ni-NTA affinity chromatography, in which sodium cholate was gradually replaced with n-Dodecyl-β-D-Maltopyranoside (DDM, Anatrace). Elution was performed using buffer containing 20 mM Tris-HCl pH 8.0, 100 mM NaCl, 1 mM MgCl$_2$, 10 µM GDP, 0.05% (w/v) DDM, 0.1 mM TCEP, and 300 mM imidazole. Further purification was achieved using a Hitrap Q column (Cytiva) equilibrated with 20 mM Tris-HCl pH 8.0, 1 mM MgCl$_2$, 10 µM GDP, and 0.03% (w/v) DDM. The purified G$_i$ heterotrimer was concentrated, flash-frozen in liquid nitrogen, and stored at −80 °C until use.

## Expression and purification of scFv16

The scFv16 construct, generously provided by Dr. Kobilka (Stanford University), was expressed and purified with minor modifications to the previously described method (Maeda et al, 2018). The construct, containing a C-terminal His$_6$ tag, was expressed in *Trichoplusia ni* (Hi5) insect cells. Cells were harvested 72 h post-infection by centrifugation, and the culture supernatant containing secreted scFv16 was incubated with Ni-NTA resin for 2 h at 4 °C. The resin was washed with buffer containing 20 mM HEPES pH 7.0, 150 mM NaCl, and 20 mM imidazole. The bound protein was eluted using the same buffer supplemented with 300 mM imidazole. The eluate was further purified by size exclusion chromatography using a column equilibrated with 20 mM HEPES pH 8.0 and 150 mM NaCl. The purified scFv16 was concentrated, flash-frozen in liquid nitrogen, and stored at −80 °C for later use.

## Expression and purification of anti-BRIL Fab

The genes encoding the anti-BRIL Fab (Fab) VH and VL domains (Tsutsumi et al, 2020) were subcloned into a pFastBac Dual vector. The Fab was expressed in *Trichoplusia ni* (Hi5) insect cells using the Bac-to-Bac expression system. Cells were harvested 72 h post-infection, and the culture supernatant containing the secreted Fab was collected. The supernatant was applied to a Ni-NTA resin pre-equilibrated with wash buffer (20 mM HEPES pH 7.5, 200 mM NaCl, and 10 mM imidazole). After washing, the Fab was eluted with buffer containing 300 mM imidazole. The eluate was further purified by size exclusion chromatography using a HiLoad 26/200 Superdex 200 column (Cytiva) equilibrated with 20 mM HEPES pH 8.0 and 150 mM NaCl. The purified Fab was collected and used for subsequent experiments.

## Expression and purification of anti-BRIL Fab nanobody

Anti-BRIL Fab nanobody (Nb) was cloned into pET-21d expression vector with a His$_6$-tag at its C-terminus. *Escherichia coli* (E. coli) Rosetta (DE3, Novagen) strain was used for expression. Nb was purified as previously described (Pardon et al, 2014). Briefly, Nb was purified using a Ni-NTA column, followed by SEC using a Superdex 200 10/300 column (Cytiva). The peak fractions were collected and stored at 4 °C until use.

## Purification of C3aR-BRIL–Fab-Nb complex and JR14a bound C3aR-BRIL–Fab-Nb complex

For the purification of the apo C3aR-BRIL-Fab-Nb complex, purified C3aR-BRIL, anti-BRIL Fab (Fab), and BRIL-Fab nanobody (Nb) were combined at a molar ratio of 1:1.2:1.5 and incubated on ice for 30 min. To remove the cleaved eGFP tag, excess Fab and Nb, the mixture was loaded onto anti-FLAG M1 affinity resin in the presence of 2 mM CaCl$_2$. The resin was washed with LMNG-containing buffer (20 mM HEPES pH 8.0, 150 mM NaCl, 20 mM imidazole, 0.005% (w/v) LMNG, 0.0005% (w/v) CHS, and 2 mM CaCl$_2$). The C3aR-BRIL-Fab-Nb complex was eluted using buffer containing 20 mM HEPES pH 8.0, 150 mM NaCl, 0.005% (w/v) LMNG, 0.0005% (w/v) CHS, 0.1 mg mL$^{-1}$ FLAG peptide, and 4 mM EDTA. Further purification was achieved by size exclusion chromatography using a Superdex 200 10/300 column (Cytiva) pre-equilibrated with 20 mM HEPES pH 8.0, 150 mM NaCl, 0.002% (w/v) LMNG, and 0.0002% (w/v) CHS. The purified complex was concentrated for cryo-EM sample preparation.

For the JR14Aa-bound C3aR-BRIL-Fab-Nb complex, 10 µM JR14a (MedChemExpress) was added to the Ni-NTA eluted C3aR-BRIL. The purification procedure was identical to that of the apo

complex, with the addition of 10 μM JR14a during all purification steps and size exclusion chromatography.

## Purification of JR14a–C3aR–G$_i$–scFv16 complex

To form the JR14a-C3aR-G$_i$-scFv16 complex, purified C3aR, human Gα$_i$ heterotrimer, and scFv16 were combined at a molar ratio of 1:1.2:1.5 in the presence of 10 μM JR14a and incubated overnight at 4 °C. The reaction mixture was then applied to an anti-GFP nanobody column (homemade) to remove excess G protein and scFv16. The column was washed with buffer containing 20 mM HEPES pH 8.0, 150 mM NaCl, 0.005% (w/v) LMNG, 0.0005% (w/v) CHS, and 10 μM JR14a. The complex was eluted by on-column cleavage with HRV 3C protease. Final purification was achieved by size exclusion chromatography using a Superdex 200 10/300 column (Cytiva) pre-equilibrated with 20 mM HEPES pH 8.0, 150 mM NaCl, 0.002% (w/v) LMNG, 0.0002% (w/v) CHS, and 10 μM JR14a. The purified complex was concentrated for cryo-EM sample preparation.

## Cryo-EM grid preparation and data collection

For cryo-EM grid preparation, 3.5 μL of purified and concentrated C3aR-BRIL–Fab-Nb complexes (apo state and JR14a-bound state) and JR14a–C3aR–G$_i$–scFv16 complex (1 mg mL$^{-1}$, 1 mg mL$^{-1}$, and 3 mg mL$^{-1}$, respectively) were applied to glow-discharged 300-mesh holey carbon grids (Quantifoil R1.2/1.3, SPI). Grids were blotted for 3 s with a blot force of 5 at 4 °C and 100% humidity, then plunge-frozen in liquid ethane using a Vitrobot Mark IV (Thermo Fisher Scientific). Initial screening was performed using a FEI Glacios microscope (Thermo Fisher Scientific) equipped with a Falcon 4 detector at the Center for Macromolecular and Cell Imaging, Seoul National University (SNU CMCI, Korea).

Data collection for the apo C3aR-BRIL–Fab-Nb complex was carried out on a Titan Krios G4 (Thermo Fisher Scientific) at the Institute of Basic Science (IBS, Korea), equipped with a K3 BioQuantum detector (Gatan). Images were acquired at a magnification of 105,000× with a calibrated pixel size of 0.848 Å. Movies were collected with a total dose of 62.6 e$^-$ Å$^{-2}$ over 54 frames per micrograph, with a defocus range of −0.9 to −1.7 μm.

The JR14a-bound C3aR-BRIL–Fab-Nb complex was imaged using a Titan Krios G4 at the Institute of Membrane Proteins (IMP, Korea), equipped with a K3 BioQuantum detector. Data were collected at a magnification of 105,000× with a calibrated pixel size of 0.85 Å. Movies were recorded with a total dose of 60.8 e$^-$ Å$^{-2}$ over 50 frames per micrograph, using a defocus range of −0.9 to −1.7 μm.

For the JR14a–C3aR–G$_i$–scFv16 complex, data were collected on a Titan Krios G4 at the Pusan National University (PNU, Korea), equipped with a Falcon 4i detector (Thermo Fisher Scientific). Images were acquired at a magnification of 96,000× with a calibrated pixel size of 0.81 Å. Movies were recorded with a total dose of 60 e$^-$ Å$^{-2}$ over 50 frames per micrograph, using a defocus range of −1.0 to −1.9 μm.

## Image processing and cryo-EM map construction

All collected image stacks were subjected to beam-induced motion correction using patch motion correction, and contrast transfer function (CTF) parameters for each non-dose-weighted micrograph were determined using patch CTF estimation in cryoSPARC v.4.5.1 (Punjani et al, 2017).

In all, 15,418 movies were processed for the C3aR-BRIL–Fab-Nb apo state complex, while 9070 movies were used for the JR14a-bound complex. Both apo and JR14a-bound complex datasets were processed similarly. Initial particle selection was performed using a Blob picker. Templates were generated through multiple rounds of two-dimensional (2D) classification and selection. The final selected particles were used for Topaz training (Bepler et al, 2019), and the extracted particles were used for 3D reconstruction. Multiple rounds of Ab-initio reconstruction and heterogeneous refinement were performed to select suitable particles, yielding 867,013 (apo) and 866,464 (JR14a-bound) particles. Further focused 3D-classification on the receptor region using RELION v.4 (Kimanius et al, 2021; Scheres, 2012) resulted in 349,007 (apo) and 342,058 (JR14a-bound) final particles. These particles were further subjected to reference-based motion correction (RBMC, cryoSPARC v.4) and CTF refinement, followed by non-uniform refinement (Punjani et al, 2020), yielding global resolutions of 2.7 Å (apo) and 3.1 Å (JR14a-bound). A mask was created for the anti-BRIL-Fab and Nb region using Chimera's Segger tool (Pettersen et al, 2004) which was used for subtraction and local refinement of the receptor region yielded final resolutions of 3.6 Å (apo) and 3.5 Å (JR14a-bound), respectively.

For the JR14a–C3aR–G$_i$–scFv16 complex, 8500 movies were processed using cryoSPARC v4.2. Particles were selected through 2D classification, and Topaz training was applied. From 557,534 Topaz-extracted particles, 372,365 were selected through 2D classification and heterogeneous refinement. These particles were further subjected to reference-based local motion correction (RBMC, cryoSPARC v.4), CTF refinement, and non-uniform refinement, yielding a global resolution of 2.53 Å. Masks for C3aR and G$_i$ heterotrimer were generated using Chimera's Segger tool. Local refinement produced maps with resolutions of 2.8 Å for C3aR and 2.42 Å for G$_i$ heterotrimer. These maps were combined using Chimera's Vop maximum tool (Pettersen et al, 2004) for model building. Detailed cryo-EM data collection parameters are provided in Appendix Table S2.

## Model building and refinement

Initial models for C3aR and C3aR-BRIL were generated using AlphaFold2 (Jumper et al, 2021). The G$_i$ heterotrimer and scFv16 structures were derived from the NPY-NPY1R-G$_i$-scFv16 structure (PDB: 7VGX). Model building and refinement were performed through iterative cycles of manual rebuilding in Coot (Emsley and Cowtan, 2004) and refinement with PHENIX (Afonine et al, 2018). The geometry of the refined structures was evaluated using MolProbity (Chen et al, 2010).

For the active JR14a–C3aR–G$_i$–scFv16 complex, the N-terminal residues (1–19), ECL2 loop (166–168 and 175–330), and the C-terminal 27 residues (456–482) of C3aR were not modeled due to poor map quality. Additionally, the α-helical region (47–193), N-terminal four residues (1–4), and some loop regions (235–249, 281–294, and 308–319) of Gα$_i$ were omitted from the final model. In the apo C3aR structure, the N-terminal 21 residues (1–21), ECL1 (86–88), ECL2 (161–330) and the C-terminal residues (440–448 and 460–482) were not modeled due to insufficient density. ICL3 (366–369) was deleted for BRIL insertion. The JR14a-bound C3aR structure had similar unmodeled regions as the apo structure, including the N-terminal 19 residues (1–19), ECL1 (86–87), ECL2 (165–169 and 175–330) and the C-tail (440–448 and 460–482). ICL3 (366–369) was also deleted for BRIL insertion.

Refinement statistics for all structures are presented in Appendix Table S2. The final models and corresponding electron density maps have been deposited in the Protein Data Bank (PDB)

and Electron Microscopy Data Bank (EMDB), respectively. All molecular graphics figures were prepared using UCSF Chimera (Pettersen et al, 2004), UCSF ChimeraX (Pettersen et al, 2021), and PyMOL (Schrödinger, 2020).

## ELISA-based surface expression assay

HEK293T cells were seeded on poly-D-lysine coated 96-well clear microplates (NEST) and transfected with pcDNA3.1 plasmid encoding C3aR or its mutants tagged with an N-terminal FLAG tag. After 48 h, cells were fixed by treatment with 4% paraformaldehyde (Tech&Innovation) and washed with PBS. After 2 h of incubation with blocking solution (5% BSA (Bovogen) in PBS), rabbit anti-FLAG antibody (Cell signaling Technology, D6W5B, 1:1000 dilution) was treated at 4 °C overnight. Cells were washed with PBS and treated with goat anti-rabbit IgG horseradish peroxidase (HRP)-conjugated antibody (Enzo Life Sciences, ADI-SAB-300-J, 1:1000 dilution) for 2 h. After washing with PBS, TMB solution (Thermo Fisher Scientific) was added for detection. When blue color appeared, 1 N HCl was added for quenching. The absorbance at 450 nm was measured using FlexStation3 microplate reader (Molecular Devices). The remaining solution was removed and Janus Green solution (0.2% w/v, TCL) was added to the cells for normalization. After excess stain was eliminated with extensive washing with distilled water, 0.1 N HCl was added for elution. The absorbance at 595 nm was measured using FlexStation3 microplate reader. The relative expression level was calculated using the ratio of absorbance at 450 nm to absorbance at 595 nm (A450/A595). The graph was plotted using GraphPad Prism 10.1.2.

## BRET assay

HEK293T cells were seeded on 6-well plates (NEST) and transfected with pcDNA3.1 plasmids encoding C3aR-eYFP, $G\alpha_i$-Rluc8, $G\beta_1$, and $G\gamma_2$ with a ratio of 5:1:1:1. After 48 h, cells were harvested and resuspended with in PBS. Cells were treated with test ligand with various concentrations and seeded on 96-well white bottom plates (SPL). After 3 min incubation at 37 °C, coelenterazine h (Nanolight) was added with a final concentration of 10 μM. Luminescence and fluorescence were measured using a Tristar 2 LB 942 multimode reader (Berthold). The BRET ratio (eYFP/Rluc8) and the difference in BRET ratio (ΔBRET) were calculated to study ligand-induced recruitment. ΔBRET was plotted as the function of concentration using a three-parameter logistic equation with GraphPad Prism 10.1.2.

## GloSensor-based cAMP assay

HEK293T cells were seeded on 6-well plates (NEST) and transfected with pGlosensor™-22F plasmids (Promega) and pcDNA3.1 plasmids encoding C3aR or its mutants an with N-terminal FLAG tag. After 48 h, cells were detached with Accutase solution (Sigma-Aldrich) and washed with PBS, and resuspended in assay buffer ($CO_2$-independent medium with 0.5 mg mL⁻¹ D-luciferin). 100,000–200,000 cells were seeded on 96-well white bottom plates (SPL) and incubated at 37 °C for 45 min followed by 45 min incubation at room temperature. After 90 min of incubation, basal luminescence signals were recorded. In total, 25 μl of test ligand with various concentrations and 1 μM Forskolin (Sigma-Aldrich) were treated to the cells and luminescence signals were measured by Tristar 2 LB 942 multimode reader (Berthold). All

data was analyzed using GraphPad Prism 10.1.2. Relative luminescence unit (%) was calculated with the signal of wells treated with vehicle as 0% and the signal of wells treated with 1 μM Forskolin as 100% (DiRaddo et al, 2014). Nonlinear regression was performed using a three-parameter logistic equation.

## Calculation of binding pocket volume

The pocket volumes for different C3aR states were measured using POVME software (Durrant et al, 2011; Durrant et al, 2014). The input for POVME was generated by defining the upper boundary of the pocket as a plane formed by the C-alpha positions of C3aR residues 82, 160, and 400.

## Statistical analysis

The statistical significance was analyzed with GraphPad prism 10.1.2 using one-way ANOVA (Mixed-effects analysis, with the Geisser–Greenhouse correction) followed by Dunnett's multiple comparisons test, with individual variances computed for each comparison (ns $P > 0.05$, *$P \leq 0.05$, **$P \leq 0.01$, ***$P \leq 0.001$, ****$P \leq 0.0001$).

# Data availability

The coordinates of the apo state C3aR, JR14a bound C3aR and JR14a-C3aR-$G_i$-scFv16 complexes have been deposited in the Protein Data Bank under accession numbers of 9ISI, 9IPY, and 9IPV, respectively. The cryo-EM density maps have been deposited in the Electron Microscopy Data Bank under accession codes EMD-60836 (C3aR focused refined map of the apo state C3aR-BRIL-Fab-Nb complex) (https://www.ebi.ac.uk/emdb/EMD-60836), EMD-60785 (JR14a and C3aR focused refined map of JR14a bound C3aR-BRIL-Fab-Nb complex) (https://www.ebi.ac.uk/emdb/EMD-60785), EMD-60782 (JR14a-C3aR-$G_i$-scFv16 global refined map), EMD-60783 ($G_i$ heterotrimer focused refined map of JR14a-C3aR-$G_i$-scFv16 complex) (https://www.ebi.ac.uk/emdb/EMD-60783), and EMD-60784(JR14a-C3aR focused refined map of JR14a-C3aR-$G_i$-scFv16 complex) (https://www.ebi.ac.uk/emdb/EMD-60784).

The source data of this paper are collected in the following database record: biostudies:S-SCDT-10_1038-S44318-025-00429-w.

# Peer review information

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

## Acknowledgements

This work was supported by the Bio&Medical Technology Development Program (No. RS-2024-00344154 to H-JC) of the National Research Foundation (NRF) and the NRF grants (No. 2022M3A9I2017587, 2023R1A2C3004205, and RS-2024-00407331 to H-JC) funded by the Korean government (MSIT). We thank the cryo-EM facilities of NEXUS consortium, supported by a NRF of Korea grant RS-2024-00440289 (to H-JC). We thank the Core Research Facilities of Pusan National University in Korea, Institute of Membrane Protein (IMP) in Korea and Dr. Bum Han Ryu at Institute for Basic Science (IBS) in Korea for supporting the cryo-EM data collection. We thank Global Science Experimental Data Hub Center (GSDC) and KREONET at Korea Institute of Science and Technology Information (KISTI) for computing resources and technical support.

## Author contributions

**Jinuk Kim**: Data curation; Validation; Investigation; Visualization; Writing—original draft. **Saebom Ko**: Data curation; Validation; Investigation; Writing—original draft. **Chulwon Choi**: Data curation; Visualization. **Jungnam Bae**: Data curation; Visualization. **Hyeonsung Byeon**: Data curation. **Chaok Seok**: Data curation. **Hee-Jung Choi**: Conceptualization; Supervision; Investigation; Writing—original draft; Writing—review and editing.

Source data underlying figure panels in this paper may have individual authorship assigned. Where available, figure panel/source data authorship is listed in the following database record: biostudies:S-SCDT-10_1038-S44318-025-00429-w.

## Disclosure and competing interests statement

The authors declare no competing interests.

# Expanded View Figures

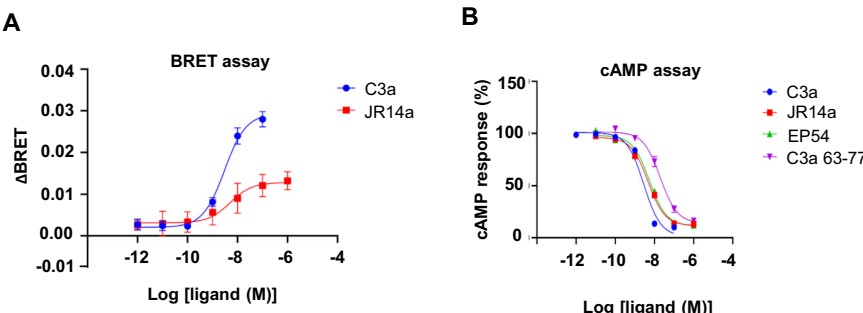

**Figure EV1. Dose-response curves of C3aR-mediated G$_i$ signaling in response to various ligands.**

(A) G$_i$ recruitment BRET assay was performed for C3a (blue circle) and JR14a (red rectangle). (B) cAMP response was assessed for C3a (blue circle), JR14a (red rectangle), EP54 (green triangle), and C3a 63–77 (purple inverted triangle). Each data point represents the mean ± standard error of the mean (S.E.M.) from $n = 4$–15 independent experiments, with detailed $n$ values provided in Appendix Table S1. Source data are available online for this figure.

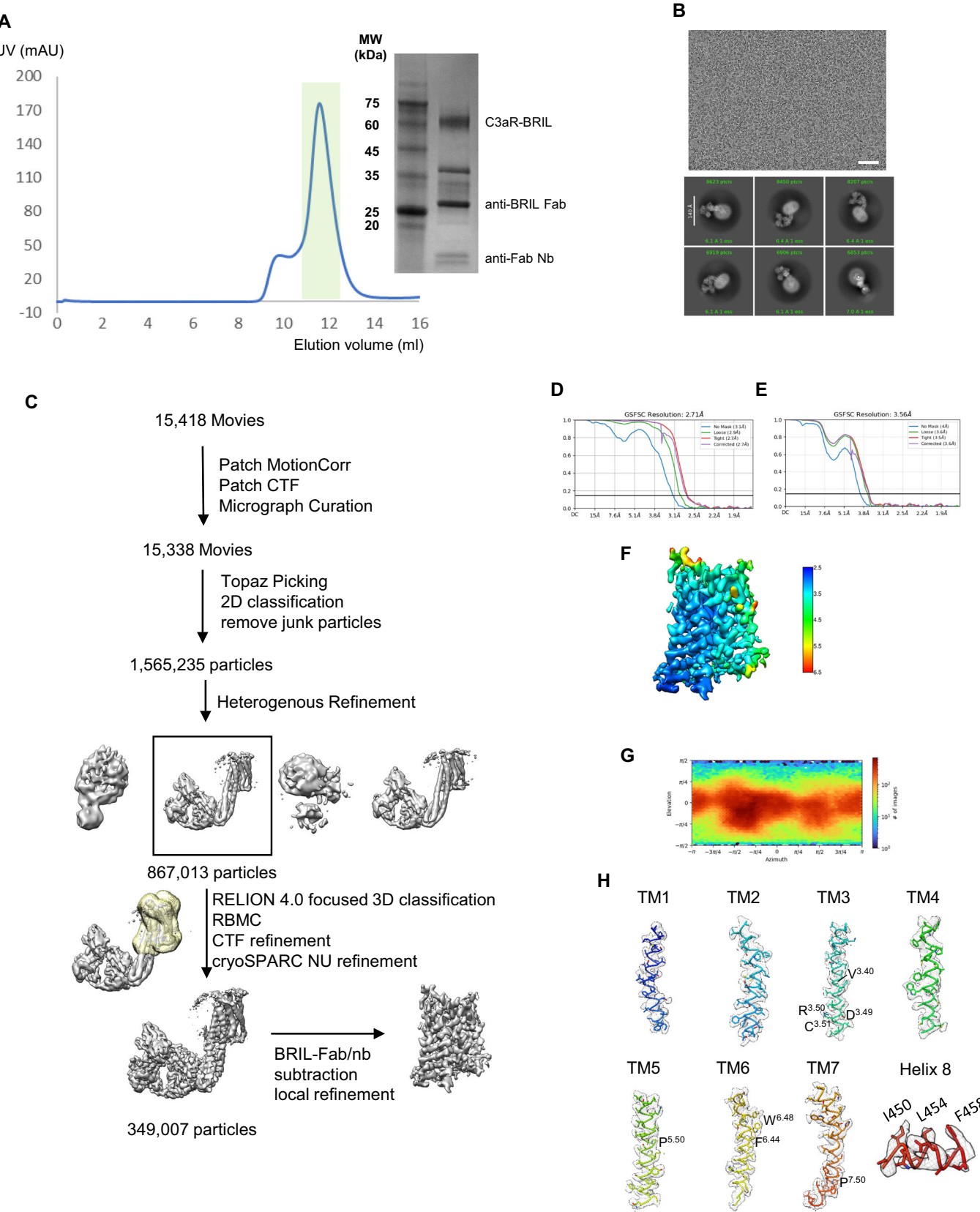

◀ **Figure EV2.  Cryo-EM data analysis of apo C3aR — BRIL with anti-BRIL Fab and anti-Fab Nanobody complex.**

(A) SEC profile (left) and SDS- PAGE (right) of purified apo C3aR—BRIL in complex with anti-BRIL Fab and anti-Fab Nb. (B) Representative micrograph (scale bar, 50 nm) and 2D average classes (scale bar, 140 Å). (C) Flowchart of data processing using cryoSPARC v4.5.1 and RELION 4.0. (D) FSC curves or cryo-EM maps for non-uniform refinement and (E) C3aR-focused local refinement. (F) Cryo-EM maps colored by local resolution of C3aR-focused local refinement. (G) The Euler angle distribution of final reconstructed local refinement map. (H) Density representation of TMs and helix8 are shown. Source data are available online for this figure.

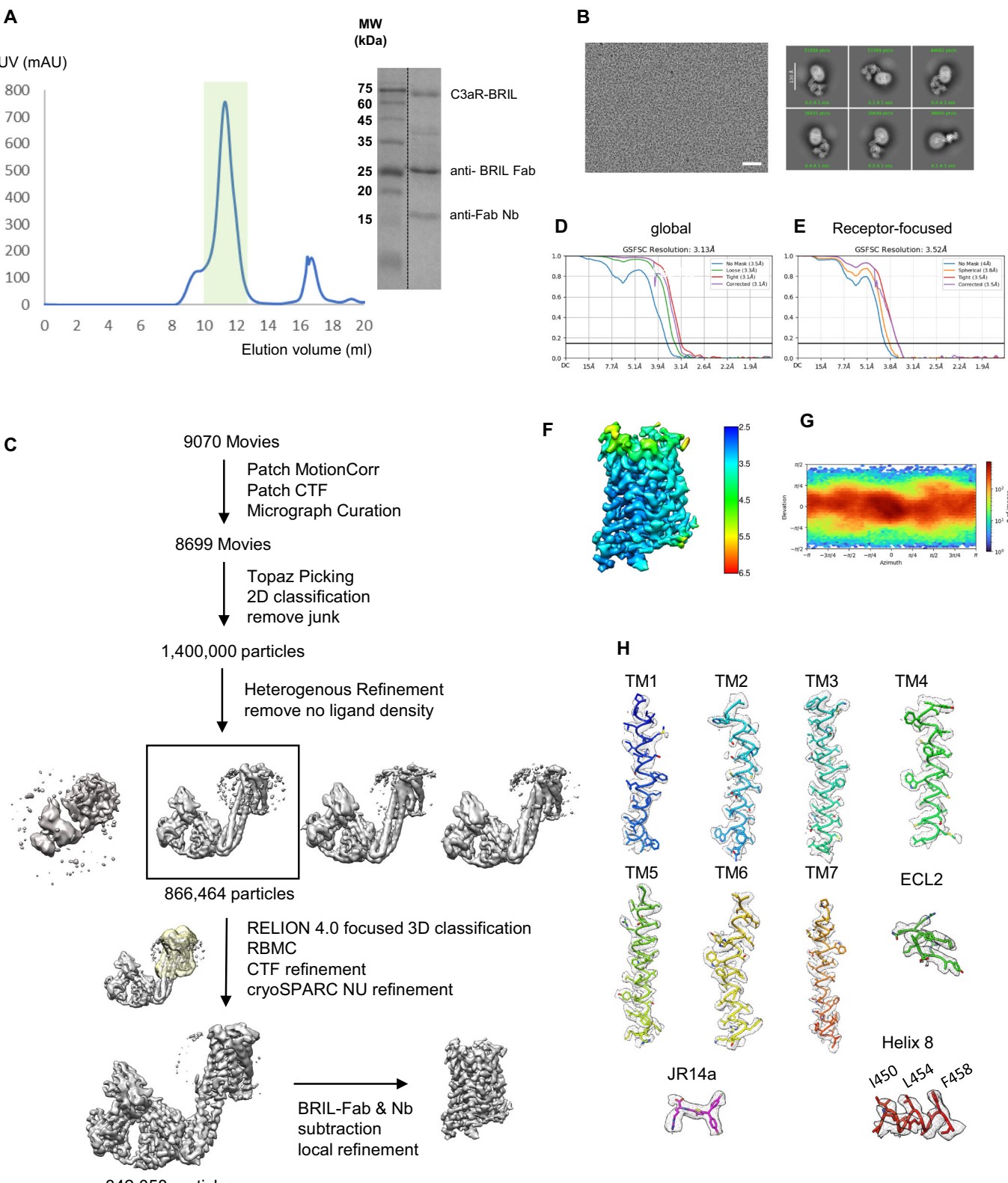

◀  **Figure EV3.   Cryo-EM data analysis of JR14a-bound C3aR-BRIL fusion with anti-BRIL Fab and anti-Fab Nanobody complex.**

(A) SEC profile (left) and SDS- PAGE (right) of purified JR14a-bound C3aR−BRIL in complex with anti-BRIL Fab and anti-Fab Nb. (B) Representative micrograph (scale bar, 50 nm) and 2D average classes (scale bar, 130 Å). (C) Flowchart of data processing using cryoSPARC v4.5.1 and RELION 4.0. (D) FSC curves for cryo-EM maps for non-uniform refinement and (E) C3aR-focused local refinement. (F) Cryo-EM maps colored by local resolution of C3aR-focused local refinement. (G) The Euler angle distribution of final reconstructed local refinement map. (H) Density representation of TMs, ECL2, JR14a and helix8 are shown. Source data are available online for this figure.

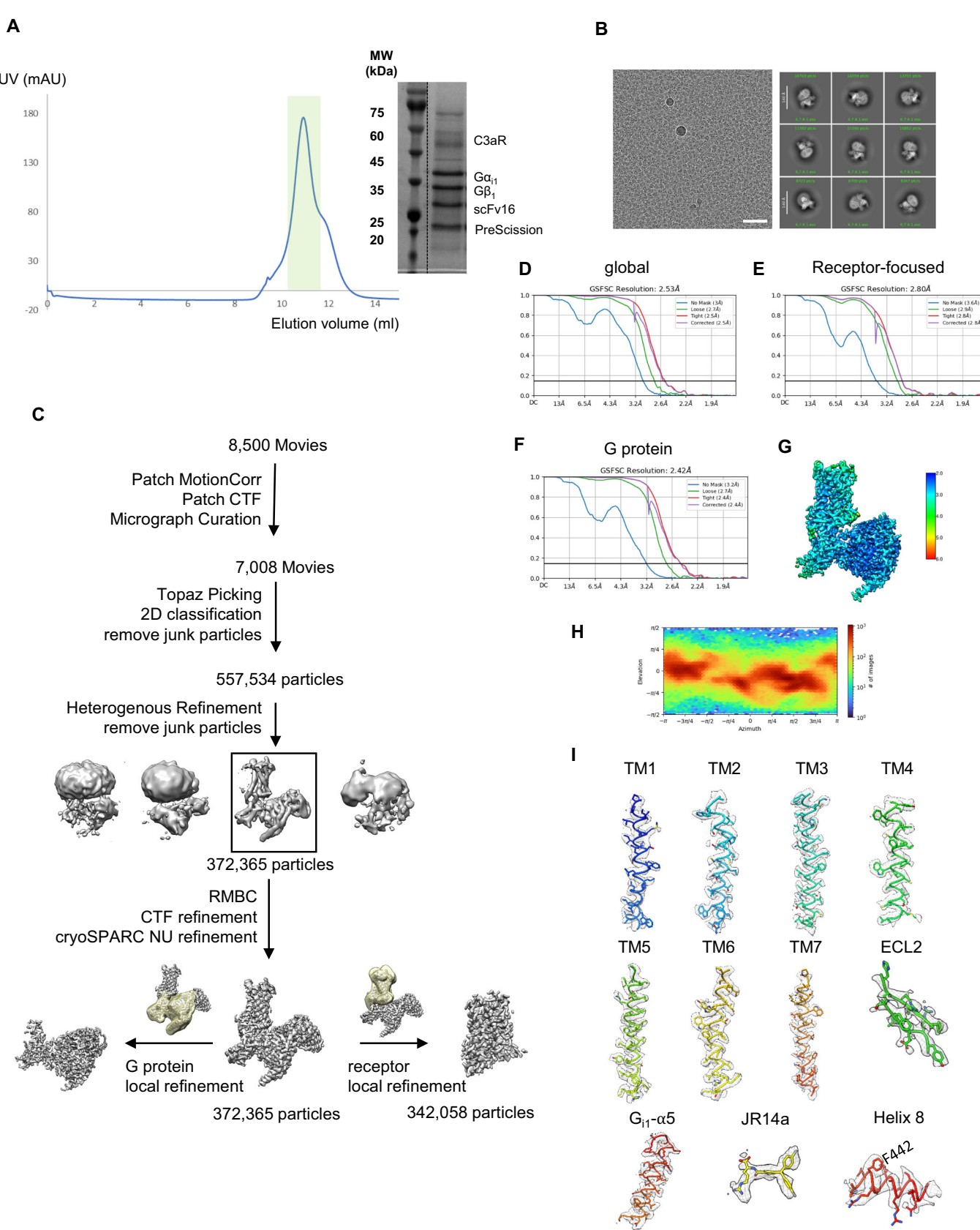

◀ **Figure EV4. Cryo-EM data analysis of JR14a-bound C3aR – G$_i$–scFv16 complex.**

(A) SEC profile (left) and SDS- PAGE (right) of purified JR14a-bound C3aR – G$_i$–scFv16 complex. (B) Representative micrograph (scale bar, 50 nm) and 2D average classes (scale bar, 140 Å). (C) Flowchart of data processing using cryoSPARC v4.5.1. (D) FSC curves for cryo-EM maps for non-uniform refinement and (E) C3aR-focused local refinement and (F) G$_i$-focused local refinement. (G) Cryo-EM maps colored by local resolution of C3aR-focused local refinement. (H) The Euler angle distribution of final reconstructed local refinement map. (I) Density representation of TMs, ECL2, α5 helix of G$_{i1}$, JR14a and helix8 are shown. Source data are available online for this figure.

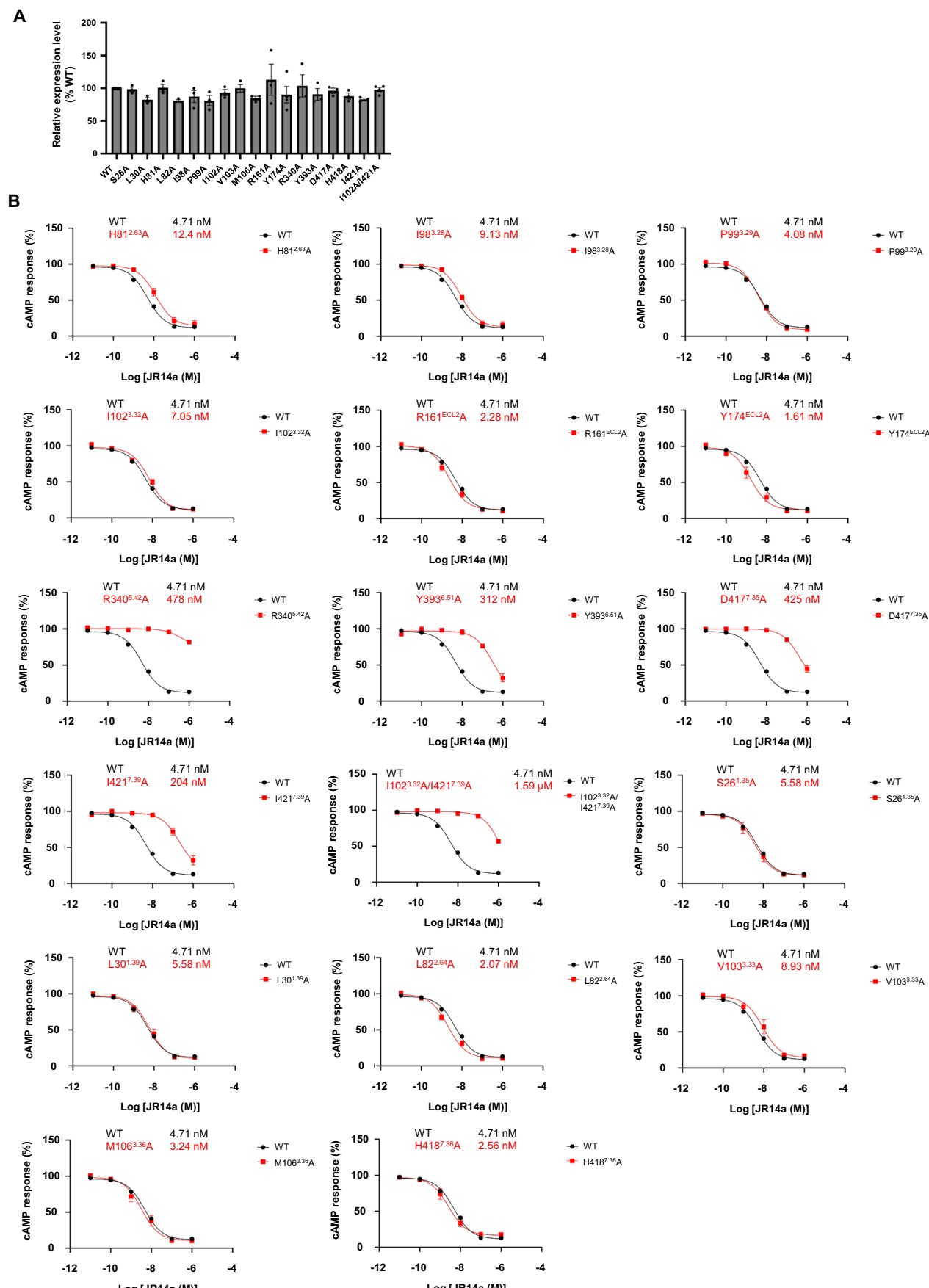

◀ **Figure EV5. Quantification of surface expression and G$_i$ signaling of C3aR mutants.**

(**A**) Surface ELISA of C3aR WT and mutants. For each data, bars and error bars indicate the means and the standard errors of the mean (S.E.M.) of 3–4 independent experiments, respectively. (**B**) cAMP response of C3aR mutants. For each C3aR mutant, relative cAMP response (%) and IC$_{50}$ was calculated and compared with WT using GraphPad Prism 10.1.2. For each data, points and error bars indicate the means and the standard errors of the mean (S.E.M.) of 3–4 independent experiments, respectively. Source data are available online for this figure.

