## [Peer Review File · The EMBO Journal]

Structural insights into small-molecule agonist recognition and activation of complement receptor C3aR

Jinuk Kim, Saebom Ko, Chulwon Choi, Jungnam Bae, Hyeonsung Byeon, Chaok Seok, and Hee-Jung Choi

Corresponding author(s): Hee-Jung Choi (choihj@snu.ac.kr)

Review Timeline:

Submission Date:	2nd Oct 24
Editorial Decision:	22nd Nov 24
Revision Received:	18th Feb 25
Editorial Decision:	5th Mar 25
Revision Received:	7th Mar 25
Accepted:	7th Mar 25

Editor: Ioannis Papaioannou

Transaction Report:

Dear Prof. Choi,

Thank you for submitting your manuscript EMBOJ-2024-119203 for consideration by The EMBO Journal, and for your patience during peer review. Your manuscript has now been seen by three experts in the field, and we have received the full set of their comments, which are included below.

As you will see, the referees recognize that the results presented in your manuscript are potentially interesting and significant, and the advance provided over the literature considerable. They also raise, however, a few technical and other concerns, and they list a number of constructive suggestions for further improvement of the study and the manuscript. They also point out that the presentation as well as the discussion of the results in the manuscript should be improved.

Given the referees' positive comments and recommendations, I would like to invite you to submit a revised version of the manuscript along with a detailed point-by-point response addressing all referees' comments. I should add that it is The EMBO Journal policy to allow only a single round of major revision, and acceptance of your manuscript will therefore depend on the completeness of your responses in this revised version. Please let me know if you have any questions or comments that you would like to discuss with me.

We generally allow three months as standard revision time (February 21, 2025). As a matter of policy, competing manuscripts published during this period will not negatively impact our assessment of the conceptual advance presented by your study. However, we request that you contact us as soon as possible upon publication of any related work, to discuss how to proceed. Should you foresee a problem in meeting this three-month deadline, please let us know in advance and we may be able to grant an extension.

Thank you for the opportunity to consider your work for publication in The EMBO Journal. I look forward to your revision.

Best regards,

Ioannis

Instructions for preparing your revised manuscript

1. When you are ready to submit the revision, please upload:

- A Word file of the manuscript text (including legends of main Figures, EV Figures and Tables). Please make sure that changes are highlighted (or "tracked") to be clearly visible.

- Individual production-quality figure files (one file per figure). When assembling your figures, please refer to our figure preparation guidelines in order to ensure proper formatting and readability in print as well as on screen:

If the data shown in a figure are obtained from n {less than or equal to} 2, please use scatter plots showing the individual data points.

- i. the name of the statistical test used to generate error bars and P values
- ii. the number (n) of independent experiments (please specify technical or biological replicates) underlying each data point (discussion of statistical methodology can be reported in the Materials and Methods section, but figure legends should contain a basic description of n , P , and the test applied)
- iii. the nature of the bars and error bars (s.d., s.e.m.).

- A point-by-point response to the referees' comments, with a detailed description of the changes made (as a word file). All referees' concerns must be fully addressed and their suggestions taken on board. When preparing your letter of response to the referees' comments, please bear in mind that this will form part of the Review Process File and will therefore be available online

to the community. Please note that you have the possibility to opt out of the transparent process at any stage prior to publication by letting the editorial office know (contact@embojournal.org); if you do opt out, the Review Process File link will point to the following statement: "No Review Process File is available with this article, as the authors have chosen not to make the review process public in this case.". For more details on our Transparent Editorial Process, please visit our website: <https://www.embopress.org/page/journal/14602075/authorguide#transparentprocess>

- Expanded View (EV) files (replacing Supplementary Information) that are collapsible/expandable online. A maximum of 5 EV Figures can be typeset. EV Figures should be cited as "Figure EV1, Figure EV2" etc. in the text, and their respective legends should be included in the manuscript file after the legends of regular figures. See detailed instructions regarding Expanded View files here:

- For the figures that you do NOT wish to display as Expanded View figures, they should be bundled together with their legends in a single PDF file called "Appendix", which should start with a short Table of Contents (including page numbers). Appendix figures should be referred to in the main text as: "Appendix Figure S1, Appendix Figure S2" etc. Please see detailed instructions here: <https://www.embopress.org/page/journal/14602075/authorguide#expandedview>

- A complete author checklist, which you can download from our author guidelines (<https://www.embopress.org/page/journal/14602075/authorguide>). Please note that the checklist will also be part of the Review Process File.

2. Please note that no statistics should be calculated and shown in Figures if $n=2$. Please also note that each p value should be reported as an exact value.

3. Before submitting your revision, primary datasets (and computer code, where appropriate) produced in this study need to be deposited in appropriate public databases (see <https://www.embopress.org/page/journal/14602075/authorguide#dataavailability>).

*** All links should resolve to a page where the data can be accessed. ***

*** Please remember to provide in the Data availability section of your revised manuscript reviewer passwords if the datasets are not yet public. ***

*** The Data Availability Section is restricted to new primary data that are part of this study. In case you have no data that require deposition in a public database, please state so instead of referring to the database: "Our study includes no data deposited in public repositories." under the heading "Data availability". ***

4. Please check that the title and the abstract of the manuscript are brief, yet explicit, even to non-specialists. The length of the title should not exceed 100 characters, and the abstract should be a single paragraph not exceeding 175 words.

5. Please also note our reference format: <https://www.embopress.org/page/journal/14602075/authorguide#referencesformat>.

7. Please remember: digital image enhancement is acceptable practice, as long as it accurately represents the original data and conforms to community standards. If a figure has been subjected to significant electronic manipulation, this must be noted in the figure legend or in the "Materials and Methods" section. The editors reserve the right to request original versions of figures and the original images that were used to assemble the figure.

8. Our journal encourages inclusion of data citations in the reference list to directly cite datasets that were obtained from public databases. Data citations in the article text are distinct from normal bibliographical citations and should directly link to the database records from which the data can be accessed. In the main text, data citations are formatted as follows: "Data ref: Smith et al, 2001" or "Data ref: NCBI Sequence Read Archive PRJNA342805, 2017". In the Reference list, data citations must be labeled with "[DATASET]". A data reference must provide the database name, accession number/identifiers, and a resolvable link to the landing page from which the data can be accessed at the end of the reference. Further instructions are available at: <https://www.embopress.org/page/journal/14602075/authorguide#referencesformat>.

9. We request authors to consider both actual and perceived competing interests. Please review our policy (<https://www.embopress.org/page/journal/14602075/authorguide#conflictofinterest>) and update your competing interests statement if necessary. Please name this section 'Disclosure and competing interests statement' and place it after the Acknowledgements section.

10. Please note that all corresponding authors are required to provide an ORCID ID upon submission of a revised manuscript

(<https://orcid.org/>). Please find instructions on how to link your ORCID ID to your account in our manuscript tracking system in our Author guidelines (<https://www.embopress.org/page/journal/14602075/authorguide#authorshipguidelines>).

11. We use CRediT to specify the contributions of each author in the journal submission system. CRediT replaces the author contribution section, which should be removed from the manuscript. Please use the free text box to provide more detailed descriptions. See also guide to authors: <https://www.embopress.org/page/journal/14602075/authorguide#authorshipguidelines>.

13. We would also welcome the submission of cover suggestions or motifs to be used by our Graphics Illustrator in designing a cover.

14. Please use the link below to submit your revision:
<https://emboj.msubmit.net/cgi-bin/main.plex>

Referee #1:

Kim et al have determined three cryo-EM structures of the C3a receptor in three different conformations, a ligand-free state, the agonist-bound receptor (no G protein) and an agonist-bound G protein-coupled state. They used the same ligand (JR14a) in both ligand-bound states, so there can be a direct comparison between the structures to define the effects of the G protein. The authors have performed a nice comparison of the structures, which is synthesised into a plausible mechanism of activation. It is fascinating to see the blocking of the G protein coupling site by H8 and its release upon receptor activation. The authors also compare the orthosteric binding pocket when bound to native peptide and JR14a, and the striking similarity is suggestive of why the ligand functions as a partial agonist rather than an inverse agonist.

The manuscript is clearly written following the route in how the research was performed and is well supported by clear figures. The inactive state structure of C3aR is an important addition to the structural repertoire of this receptor as it provides a clear anchor for understanding receptor activation and also as a foundation for the development of inverse agonists. There are few points where clarification would enhance the manuscript.

Major points.

1. The authors have decided to write the manuscript as the research unfolded, starting with the agonist-bound intermediate structure, then going to the active G-protein coupled state and finally the inactive state. This makes the manuscript disjointed and not so easy to follow. It would be far easier for the reader if the three structures are presented in the first section. This would then allow the conclusion that the intermediate is an inactive-like state obvious. The discussion then can be broken down logically into, for example, (i) the overall architecture differences eg. H8, outward movement of H6 and inward movement of H7, (ii) the orthosteric binding pocket differences, (iii) the mechanism of activation. Having all three structures present in each alignment will be possible if the authors use a narrower width of helix in their cartoons (e.g. in Pymol : cartoon_oval_width 0.6 rather than 1.35)
2. It would be helpful for the reader to state numbers when comparing receptor similarity and distances moved e.g. the rmsd between the ligand-free state and the intermediate and active state is 0.5 Å and 1.7 Å respectively, and the outward movement of H6 is 5 Å whilst H7 moves inwards by 5 Å. The reader can then immediately see the differences to other Class A receptors (limited movement of H6 and large movement of H7).
3. Is there a significant difference in the volume of the orthosteric binding pocket in the different states? In the beta1-adrenoceptor there is a clear allosteric effect of G protein coupling on the orthosteric binding pocket that results in increased ligand affinity due to the contraction of the ligand binding pocket (10.1126/science.aau5595). As you have used the same ligand in an inactive-like state and the active state it would be informative to do the comparison and discuss in comparison to beta1.
- 4.

Minor points

1. Quoted numbers should have the number of decimal places appropriate to the measurement. For example, resolution quoted in the main text should be to one decimal place as the error for structures at about 3 Å resolution is probably of the order of ~0.3 Å. Similarly, in Supplementary Table 2 there is no justification for three decimal places when the errors are up to 23.
2. In Supplementary Figures 1, 4 and 8 in panel (c) and (f) the final structure looks very noisy so could be contoured differently to show the helices clearly. In addition, the colour scale should be modified so that the local resolution map is not just

predominantly one colour e.g. {plus minus} 1A of the resolution.

3. Page 6, para 3: It is better to define the agonist-bound state as an intermediate based on a comparison with the ligand-free state and the G protein coupled state and the position of conserved microswitches within the structure such as the ionic lock, NPxxY, PIF etc. This is because in some cases an agonist-bound intermediate state (without G protein) can be 90% of the way towards a fully active state; see for example agonist bound states of the adenosine A2a receptor and Ste2.
4. Page 7 para 2, line 2; please show a diagram showing the differences in contacts between the native ligand and JR14a in the orthosteric binding pocket.
5. Although the manuscript is very well written, there are many places where 'a' and 'the' should be inserted, so running this through a grammar checker would be useful.

Referee #2:

The manuscript by Kim et. al. reports the cryo-EM structures of C3aR with modest resolutions compared to the published literature. The authors have also performed expression profiles of native and mutant C3aR and cellular signaling studies to support their structural data. Furthermore, the authors have tried to hypothesize a mechanism of activation of C3aR in response to a small molecule ligand JR14a based on the cryo-EM data in the light of C5aR. From structural biology point of view, the manuscript provides important insight how JR14a interacts with C3aR in reference to its native ligand C3a. My comments are as follows:

1. The authors have obtained the structure of inactive, meta-active, and fully active C3aR (one of the complement system GPCRs involved in various diseases) with a small molecule ligand JR14a derived from SB290157 (suggested to be a partial agonist earlier, <https://doi.org/10.3389/fphar.2020.591398>). The methods and techniques used for structural studies appear to be adequate. The refined modeled structures based on the information provided in Sup table 1 appear to be good.
2. However, the ECS (especially the ECL2) is not properly resolved (compared to reported 8i95 and 8hk3) in any of the structures in the current study. Authors should provide a comparative statement indicating what could have contributed to this observation.
3. Authors should include a statement indicating what could be the reasons behind the contrasting results (antagonist vs. partial agonist) of JR14a. The consistency of the results across different tissue types should also be considered or discussed.
4. Between C3a and C5a, the later one is a more potent pro-inflammatory molecule in the end stage of the complement cascade involved in various diseases. Therefore, from a disease model point of view, a more recent review on complement system, (e.g. doi: 10.1016/j.intimp.2023.110081) should be incorporated in addition to the C3a in the introduction section.
5. The data presented in Fig 1c (left panel) indicates that JR14a could be a partial agonist (the right term to use for JR14a) in reference to C3a (considering the scale of biological response), whereas there is no such difference in cAMP profile. What could contribute to a similar level of secondary messenger stimulation by JR14a while it does not recruit the G-protein efficiently like C3a.
6. The authors should clarify how surface expression is related to structural stability of C3aR? Furthermore, Fig. S3 indicates that JR14a makes strong contact with S26, Y393, R340, Y174, R161 of C3aR. However, mutation-based cell signaling data presented in Fig. S5 indicates that except for Y393, and R340, mutations of other residues have no significant effect on cAMP production in response to JR14a. Moreover, data presented here <https://doi.org/10.1038/s41589-023-01339-w> suggests that R161A decreases the potency of C3a ~1000 fold toward C3aR. Conversely, R161A makes no such impact on JR14a. Authors should rationalize this observation.
7. Further, authors should take note and try to acknowledge that the conformational variants observed for C3aR in response to JR14a has also been hypothesized for C5aR previously (doi: 10.1080/07391102.2024.2305698)
8. Though the authors talk about the role of S26, L30, and L82 of C3aR toward JR14a binding, it is not clear why the data related to S26A mutant is not provided, especially when S26 is shown to interact with JR14a in Fig. S3, though it is not clear what type of interactions is possible between the carbonyl oxygen and chlorine group of JR14a.
9. In Fig. 6 authors emphasize on H8 orientation in C3aR activation in response to ligand binding. However, authors does not discuss why the length of H8 keeps changing from structure to structure especially when the full-length WT C3aR is expressed and H8 is stabilized by a network of hydrophobic interactions. For instance, in JR14a-C3aR-Gi-scFv16, the H8 is of 16 aa, whereas in apo-C3aR and meta-active C5aR it is of 9 aa. Conversely, in 8i95, the H8 is of 11 aa, and in 8hk3, it is of 16 aa. Is there a possibility for ligand bound C3aR (even other GPCRs) to interact with adapter proteins without altering the H8 orientation?
10. So far as activation switch is concerned, authors should provide signaling data (basal activity) for double mutant (I102A,

I421A) both in the presence and absence of C3a (JR14a) and for possible generalization of the proposed mechanism. Notably, I102A (93% relative expression) and I421A (~83%) does affect the expression level of C3aR differently; however, only I421A produces dramatic change in cAMP production in response to JR14a and not I102A. Interestingly, I102A or I421A does not affect C3a signaling (Sup table 2).

11. Minor comment: try replacing "constitutional activity" by "constitutive activity"

Referee #3:

This manuscript reported the cryo-EM structures of C3aR in complex with a small molecule synthetic agonist JR14a in two distinct conformational states: an intermediate state and a fully active Gi-bound state. Also, the authors presented the apo structure of C3aR and offered insights into its inactive conformation, which remain elusive till now. Comparison of the apo, intermediate, and fully active structures provide a comprehensive understanding of the activation mechanism of C3aR. This paper advances the field of structure-based drug design by providing valuable insights into the conformational landscape of C3aR. These information are significant to the field. The paper is organised well overall, and the results are presented with clarity and substantiated by strong analytical techniques. Therefore this manuscript is recommended for the publication in The EMBO Journal.

Comments:

"Recent structural studies of C3aR and C5aR1 have greatly advanced our understanding of anaphylatoxin recognition by their receptors. The active structures of C5a-bound C5aR1 coupled to Gi and the inactive structure of antagonist-bound C5aR1 have revealed specific ligand recognition and the conformational changes associated with C5aR1 activation¹⁰⁻¹³."

Comment: recent research of C5aR should be updated.

J Struct Biol. 2024 Sep;216(3):108117.

"Our findings provide valuable insights for structure-based drug design targeting C3aR..."

The authors should discuss more about how the data presented could be informative to develop C3aR-targeting drugs.

We thank the reviewers for their suggestions and comments.

We greatly appreciate their time and efforts in reviewing our manuscript. We believe that our revised manuscript has benefited from their insightful suggestions.

We have addressed every point raised by the reviewers in detailed point-by-point responses (written in blue),

REVIEWER COMMENTS

Referee #1:

Kim et al have determined three cryo-EM structures of the C3a receptor in three different conformations, a ligand-free state, the agonist-bound receptor (no G protein) and an agonist-bound G protein-coupled state. They used the same ligand (JR14a) in both ligand-bound states, so there can be a direct comparison between the structures to define the effects of the G protein. The authors have performed a nice comparison of the structures, which is synthesised into a plausible mechanism of activation. It is fascinating to see the blocking of the G protein coupling site by H8 and its release upon receptor activation. The authors also compare the orthosteric binding pocket when bound to native peptide and JR14a, and the striking similarity is suggestive of why the ligand functions as a partial agonist rather than an inverse agonist.

The manuscript is clearly written following the route in how the research was performed and is well supported by clear figures. The inactive state structure of C3aR is an important addition to the structural repertoire of this receptor as it provides a clear anchor for understanding receptor activation and also as a foundation for the development of inverse agonists. There are few points where clarification would enhance the manuscript.

We appreciate the reviewer's comments on our research findings.

Major points.

1. The authors have decided to write the manuscript as the research unfolded, starting with the agonist-bound intermediate structure, then going to the active G-protein coupled state and finally the inactive state. This makes the manuscript disjointed and not so easy to follow. It would be far easier for the reader if the three structures are presented in the first section. This would then allow the conclusion that the intermediate is an inactive-like state obvious. The discussion then can be broken down logically into, for example, (i) the overall architecture differences eg. H8, outward movement of H6 and inward movement of H7, (ii) the orthosteric binding pocket differences, (iii) the mechanism of activation. Having all three structures present in each alignment will be possible if the authors use a narrower width of helix in their cartoons (e.g. in Pymol : cartoon_oval_width 0.6 rather than 1.35)

We thank the reviewer for the suggestion. In this revised manuscript, we reorganized the results section as suggested by the reviewer. The result topic orders are as below;

- Apo, intermediate, and fully active structures of C3aR (comparison of overall architectures here)
- Comparison of JR14a binding pockets in the intermediate and active states
- Comparison of binding modes of JR14a, EP54, and C3a
- Molecular basis for basal activity of C3aR

- Activation mechanism of C3aR upon JR14a binding

Also, we have included the alignment of all three structures in Figure 1D (new figure).

2. It would be helpful for the reader to state numbers when comparing receptor similarity and distances moved e.g. the rmsd between the ligand-free state and the intermediate and active state is 0.5 Å and 1.7 Å respectively, and the outward movement of H6 is 5 Å whilst H7 moves inwards by 5 Å. The reader can then immediately see the differences to other Class A receptors (limited movement of H6 and large movement of H7).

We appreciate the reviewer's suggestion. We have included the RMSD values between the apo and the intermediate, and between the apo and active state as 0.5 Å and 1.7 Å respectively, in the main text. Also, we state that there are 5 Å outward and 5 Å inward movements of TM6 and TM7, respectively, in the results section.

Pages 6-7, lines 116-121,

"The structure of the JR14a-C3aR-Gi complex exhibits typical features of active structure of class A GPCRs, including a 5 Å outward movement of TM6 and a 5 Å inward shift of TM7 at the cytoplasmic side, compared to the apo structure (RMSD = 1.7 Å for 206 Cα atoms) (Fig 1D). In contrast, JR14a-bound C3aR-BRIL shows high structural similarity to apo C3aR-BRIL, with an RMSD of 0.5 Å for 224 Cα atoms, adopting an inactive conformation"

3. Is there a significant difference in the volume of the orthosteric binding pocket in the different states? In the beta1-adrenoceptor there is a clear allosteric effect of G protein coupling on the orthosteric binding pocket that results in increased ligand affinity due to the contraction of the ligand binding pocket (10.1126/science.aau5595). As you have used the same ligand in an inactive-like state and the active state it would be informative to do the comparison and discuss in comparison to beta1.

We thank the reviewer for the suggestion. Following your recommendation, we performed the volume calculation using the same method as in the β1-adrenoceptor study (10.1126/science.aau5595). Consistent with the reported allosteric effect of G protein coupling, we observed a contraction of the ligand-binding pocket upon activation (647 Å³ for intermediate state vs. 575 Å³ for active state). This data has been included in the results section and the corresponding figure has been provided in Appendix Figure S2.

Pages 8-9, lines 170-178,

"To assess whether these conformational differences affect the ligand-binding pocket volume, we calculated the pocket volume using a previously reported method for β1AR ligand binding studies (Warne et al, 2019). Our results indicate that the JR14a-bound intermediate state has a larger ligand-binding pocket (647 Å³) than the G protein-bound active structure (575 Å³). This decrease in the volume of the ligand-binding site in the active state is likely driven by an allosteric effect from G protein coupling, which induces conformational changes such as the upward rotamer shift of M106 (Appendix Fig S2). These findings align with β1AR study, which showed a correlation between a decrease in the volume of the orthosteric binding pocket and an increase in agonist-binding affinity in the active state (Warne et al., 2019)."

Minor points

1. Quoted numbers should have the number of decimal places appropriate to the measurement. For example, resolution quoted in the main text should be to one decimal place as the error for structures at about 3 Å resolution is probably of the order of ~0.3 Å. Similarly, in Supplementary Table 2 there is no justification for three decimal places when the errors are up to 23.

We thank the reviewer for the comment. As suggested by the reviewer, we have corrected the numerical precision.

Page 4, lines 82-84,

“We determined the structures of the ligand-free and G protein-free apo state of C3aR at 3.6 Å resolution, the JR14a-bound C3aR in the absence of G protein at 3.5 Å resolution, and the JR14a-C3aR-G protein complex at 2.5 Å resolution.”

Also, corrected in “Appendix Table S2”.

2. In Supplementary Figures 1, 4 and 8 in panel (c) and (f) the final structure looks very noisy so could be contoured differently to show the helices clearly. In addition, the colour scale should be modified so that the local resolution map is not just predominantly one colour e.g. {plus minus} 1Å of the resolution.

We thank the reviewer for the comment. We have replaced the cryo-EM figures and local resolution maps as recommended. These updates are reflected in Fig EV 2, 3, and 4.

3. Page 6, para 3: It is better to define the agonist-bound state as an intermediate based on a comparison with the ligand-free state and the G protein coupled state and the position of conserved microswitches within the structure such as the ionic lock, NPxxY, PIF etc. This is because in some cases an agonist-bound intermediate state (without G protein) can be 90% of the way towards a fully active state; see for example agonist bound states of the adenosine A2a receptor and Ste2.

We thank the reviewer for the suggestion. We compared the conserved motifs, the ionic lock, PIF and NPxxY, known for undergoing conformational changes during activation as shown in Figure 1E (new figure). Unlike the case the reviewer mentioned that an agonist-bound intermediate state (without G protein) can be 90% of the way towards a fully active state as in adenosine A2a receptor, our JR14a-bound C3aR (without G protein) represents more inactive-like conformations, although JR14a binding induced the local conformational changes on the extracellular sides of TMD and ECL2 of C3aR. Therefore, we define our JR14a-bound C3aR without G protein as an early intermediate state, engaging local changes on an agonist binding pocket but not propagated into the conserved microswitch regions in the TMD. We have added this information in the manuscript.

Page 7, lines 121-124

“This inactive state is further supported by the conformation of conserved microswitches, such as the DRY, PIF, and NPxxY motifs, as well as the toggle switch, all of which exhibit an inactive-like configuration in both the JR14a-bound C3aR-BRIL and apo C3aR-BRIL structures (Fig 1E).”

4. Page 7 para 2, line 2; please show a diagram showing the differences in contacts between the native ligand and JR14a in the orthosteric binding pocket.

We thank the reviewer for the suggestion. We have included a diagram showing the differences in contacts between C3a and JR14a in the ligand binding pocket in Figure 3B (new figure).

5. Although the manuscript is very well written, there are many places where 'a' and 'the' should be inserted, so running this through a grammar checker would be useful.

We thank the reviewer's comment. As requested by the reviewer, we have had the revised manuscript professionally proofread to thoroughly check the grammar (Premium Editing Service by Editage).

Referee #2:

The manuscript by Kim et. al. reports the cryo-EM structures of C3aR with modest resolutions compared to the published literature. The authors have also performed expression profiles of native and mutant C3aR and cellular signaling studies to support their structural data. Furthermore, the authors have tried to hypothesize a mechanism of activation of C3aR in response to a small molecule ligand JR14a based on the cryo-EM data in the light of C5aR. From structural biology point of view, the manuscript provides important insight how JR14a interacts with C3aR in reference to its native ligand C3a. My comments are as follows:

1. The authors have obtained the structure of inactive, meta-active, and fully active C3aR (one of the complement system GPCRs involved in various diseases) with a small molecule ligand JR14a derived from SB290157 (suggested to be a partial agonist earlier, <https://doi.org/10.3389/fphar.2020.591398>). The methods and techniques used for structural studies appear to be adequate. The refined modeled structures based on the information provided in Sup table 1 appear to be good.

We appreciate the reviewer's comments on our research findings.

2. However, the ECS (especially the ECL2) is not properly resolved (compared to reported 8i95 and 8hk3) in any of the structures in the current study. Authors should provide a comparative statement indicating what could have contributed to this observation.

We thank the reviewer for the comment. In the case of EP54-C3aR-Go structure (8i95), peptide ligand EP54 stabilized ECL2 by direct interaction with it. In the case of apo-C3aR-Gi structure (8hk3), although there is no ligand for providing stable interaction with ECL2, TM2 undergoes an unusual inward movement toward the ligand-binding pocket, partially occupying the ligand-binding pocket and forming contacts with ECL2. This TM2 movement stabilizing the ECL2, has not been observed in other reported C3aR structures, including another apo C3aR structure in complex with Go (PDB: 8I9S) (see figure below).

In our study, the JR14a-bound C3aR structures (both with and without Gi protein) exhibit ECL2 density comparable to previously reported peptide ligand-bound structures, likely due to JR14a-mediated interactions with ECL2, although their interactions are not as extensive as shown in peptide ligand-bound C3aR (see figures below). In contrast, our cryo-EM map of apo structure shows poorer ECL2 density, and thus ECL2 was not modeled. A similar phenomenon was observed in the apo C3aR-Go structure (PDB: 8I9S), where ECL2 is modeled but it appears less defined in the cryo-EM map, even though this structure forms an active conformation.

These findings suggest that the resolution differences observed in the ECL2 regions primarily result from the presence or absence of stabilizing interactions with the ligand. JR14a stabilize ECL2 through direct interactions, though to a lesser extent than peptide ligands, such as C3a and EP54. In the apo state, this region remains flexible and less defined due to the lack of ligand-mediated stabilization. We have included a statement in the Results section discussing the poor ECL2 density observed in the apo structure.

Page 7, lines 141-144,

“Specifically, while a significant portion of ECL1 and ECL2 was unresolved in the apo structure, likely due to the high flexibility of these extracellular regions in the absence of a ligand, a specific region of ECL2 was well-resolved in the JR14a-C3aR-BRIL structure, by participating in the interaction with JR14a.”

3. Authors should include a statement indicating what could be the reasons behind the contrasting results (antagonist vs. partial agonist) of JR14a. The consistency of the results across different tissue types should also be considered or discussed.

We thank the reviewer for the comment. To address the contrasting reports on JR14a’s activity, we have discussed the potential role of β -arrestin-mediated internalization, which may lead to receptor depletion from the cell surface, reducing responsiveness to subsequent JR14a stimulation. JR14a-induced signaling was performed in HEK293 cells as well as in human monocyte THP-1 cells, which naturally express functional C3aR (Luo et al., 2025). This explanation has been incorporated in the Introduction section.

Page 4, lines 68-75,

“Of note, JR14a inhibited forskolin-induced cAMP production in a dose-dependent manner, not only in HEK293 cells but also in human monocyte THP-1 cells, which naturally express functional C3aR (Luo et al, 2025). Additionally, receptor desensitization following ligand-induced stimulation may explain its antagonist-like effects, as evidenced by the abolition of C3a-induced calcium signaling when C3a was treated 10 minutes after JR14a stimulation, indicating JR14a-induced receptor desensitization (Luo et al., 2025). Another study further demonstrated that SB290157 and JR14a act as a potent agonist for C3aR, with their blockade of calcium influx attributed to β -arrestin mediated internalization (Mathieu et al, 2005; Rodriguez et al, 2024).”

4. Between C3a and C5a, the later one is a more potent pro-inflammatory molecule in the end stage of the complement cascade involved in various diseases. Therefore, from a disease model point of view,

a more recent review on complement system, (e.g. doi: 10.1016/j.intimp.2023.110081) should be incorporated in addition to the C3a in the introduction section.

We appreciate the reviewer's suggestion regarding the inclusion of a broader disease model perspective. A reference to the suggested review on complement system has been added to the Introduction section.

Page 3, lines 48-50,

"As C5aR1 plays a pivotal role in the late stage of the C3 complement cascade, C5aR1 has been studied as an important drug target in a wide range of immune-related diseases (Ghosh & Rana, 2023)."

5. The data presented in Fig 1c (left panel) indicates that JR14a could be a partial agonist (the right term to use for JR14a) in reference to C3a (considering the scale of biological response), whereas there is no such difference in cAMP profile. What could contribute to a similar level of secondary messenger stimulation by JR14a while it does not recruit the G-protein efficiently like C3a.

We thank the reviewer for the comment. While BRET assay measures direct interaction between the

(Luo P et al, 2025)

two proteins, cAMP assay measures the accumulation of second messenger over periods of about 20 minutes. Although this accumulated amount of cAMP reflects G protein signaling, it is also affected by the activity of adenylyl cyclase and phosphodiesterase, and even receptor regulation such as internalization. For example, a recent study demonstrated that JR14a and C3a recruits β -arrestin2 with different efficacy and potency, inducing receptor internalization within 15 minutes. This suggests that JR14a may maintain prolonged G_i protein-mediated signaling at the plasma membrane, resulting in sustained cAMP inhibition despite less efficient G protein recruitment.

Therefore, these complicate modulations of signaling may cause the differences in profiles of G protein recruitment and changes in cAMP concentrations, which may not be directly correlated.

6. The authors should clarify how surface expression is related to structural stability of C3aR? Furthermore, Fig. S3 indicates that JR14a makes strong contact with S26, Y393, R340, Y174, R161 of C3aR. However, mutation-based cell signaling data presented in Fig. S5 indicates that except for Y393, and R340, mutations of other residues have no significant effect on cAMP production in response to JR14a. Moreover, data presented here <https://doi.org/10.1038/s41589-023-01339-w> suggests that R161A decreases the potency of C3a ~1000 fold toward C3aR. Conversely, R161A makes no such impact on JR14a. Authors should rationalize this observation.

We appreciate the reviewer's suggestion.

1) We have clarified the relationship between surface expression and structural stability of C3aR. Specifically, we added the following explanation in the manuscript.

Page 7, lines 135-138,

"As GPCRs that are incompletely folded or misfolded fail to pass the ER quality control mechanism and are marked for degradation (Dong et al, 2007), a significantly low level of surface expression could indicate that GPCRs are structurally unstable."

2) Regarding the reviewer's comment that the S26A, R161A, and Y174A mutants have no significant effect on G_i signaling in response to JR14a, our structural analysis provides insight into this observation.

In the fully active C3aR structure, JR14a binds deeply within the TMD pocket, increasing its distance from S26 (4.3 Å), suggesting that S26 plays a minimal role in JR14a binding. Meanwhile, R161 and Y174, located in the ECL2, interact with JR14a within a 3–4 Å range; however, these interactions appear relatively weak and transient due to the inherent flexibility of ECL2. In contrast, in the C3a-bound active structure, R161 and Y174 form extensive stabilizing contacts with the C-terminal carboxyl group, the carbonyl backbone, and L73 of C3a (ionic, polar and van der Waals interactions), positioning them correctly for interaction. We have included this analysis in the manuscript along with a supporting figure (Appendix Fig S5).

Page 10, lines 200-206,

“However, the R161^{ECL2A} and Y174^{ECL2A} mutants respond differently: signaling was greatly reduced for C3a but not for JR14a (Appendix Fig S3, S4 and Appendix Table S1). This can be attributed to the stable contacts between R161^{ECL2} and Y174^{ECL2} of C3aR and the C-terminal carboxyl group as well as the L75 carbonyl group and L73 of C3a, which correctly position these residues for interaction with C3a. In contrast, in the JR14a-bound structure, R161^{ECL2} and Y174^{ECL2} lack these additional stabilizing interactions with JR14a, making their contacts more transient due to ECL2’s inherent flexibility (Appendix Fig S5).”

7. Further, authors should take note and try to acknowledge that the conformational variants observed for C3aR in response to JR14a has also been hypothesized for C5aR previously (doi: 10.1080/07391102.2024.2305698)

We thank the reviewer for the comment. We thank the reviewer for pointing out the relevant work on C5aR. We acknowledge that the conformational variants observed in C3aR in response to JR14a may share similarities with those hypothesized for C5aR. We add relevant information to the manuscript where needed.

Page. 16, lines 354-356,

“Additionally, beyond the conformational states associated with G protein coupling, it is important to examine the conformational subsets linked to β-arrestin coupling, as previously studied for C5aR1 (Gupta et al, 2024).”

8. Though the authors talk about the role of S26, L30, and L82 of C3aR toward JR14a binding, it is not clear why the data related to S26A mutant is not provided, especially when S26 is shown to interact with JR14a in Fig. S3, though it is not clear what type of interactions is possible between the carbonyl oxygen and chlorine group of JR14a.

We thank the reviewer for the comment. Based on the current structural data, the interaction between S26 and JR14a varies across different conformational states. In the intermediate structure, the chlorine atom of JR14a is positioned ~2.5 Å from the hydroxyl oxygen of S26, suggesting the possibility of a dipole interaction between them. However, in the fully active structure, the chlorine atom is 4.9 Å away from the hydroxyl oxygen, while it is closer (4.3 Å) to the beta carbon (Cβ) of S26. These changes in interaction may indicate a weak and transient interaction between S26 and JR14a.

Indeed, our mutagenesis study showed that S26A mutant did not display significant effect on cAMP signaling induced by JR14a. We also examined its surface expression level and cAMP signaling induced by C3a, which had no significant difference from the wild-type.

We have incorporated the S26A mutant data in Figure 2D, Figure EV5, and Appendix Figure S4.

9. In Fig. 6 authors emphasize on H8 orientation in C3aR activation in response to ligand binding. However, authors does not discuss why the length of H8 keeps changing from structure to structure especially when the full-length WT C3aR is expressed and H8 is stabilized by a network of hydrophobic interactions. For instance, in JR14a-C3aR-Gi-scFv16, the H8 is of 16 aa, whereas in apo-C3aR and meta-active C5aR it is of 9 aa. Conversely, in 8i95, the H8 is of 11 aa, and in 8hk3, it is of 16 aa. Is there a possibility for ligand bound C3aR (even other GPCRs) to interact with adapter proteins without altering the H8 orientation?

We thank the reviewer for the comment. In the apo and intermediate states of C3aR, H8 adopts an inverted conformation spanning residues 450–458 (9 aa), stabilized by hydrophobic interactions within the receptor core. This conformation has only been previously observed in C5aR1 bound to an antagonist. Upon activation and G protein binding, H8 shifts outward, enabling G protein coupling. A similar conformational transition of H8 was observed during C5aR1 activation. In the fully active state of C3aR, H8 extends into a helix spanning residues 440–455 (16 aa), while residues 456–482 remain unresolved, suggesting a flexible loop beyond the helical region.

The differences in H8 length observed across active structures, such as 8i95 and 8HK3, are likely due to variations in cryo-EM map resolution, affecting model building. These differences appear to be methodological rather than biological. The apparent changes in H8 length across C3aR conformational states reflect structural rearrangements necessary for activation, including the transition from an inward- to outward-facing orientation and the formation of a stabilized helical structure, along with resolution limitations in cryo-EM data.

Regarding the possibility of C3aR interacting with other adapter proteins without altering H8 orientation, our structural data show that H8 physically obstructs G protein access to the receptor, making its structural rearrangement essential for G protein coupling. However, further studies are required to explore whether C3aR can interact with other signaling proteins independently of H8 repositioning.

10. So far as activation switch is concerned, authors should provide signaling data (basal activity) for double mutant (I102A, I421A) both in the presence and absence of C3a (JR14a) and for possible generalization of the proposed mechanism. Notably, I102A (93% relative expression) and I421A (~83%) does affect the expression level of C3aR differently; however, only I421A produces dramatic change in cAMP production in response to JR14a and not I102A. Interestingly, I102A or I421A does not affect C3a signaling (Sup table 2).

We thank the reviewer's comment. The difference in surface expression levels between the I102A and I421A mutants is not substantial enough to affect the results of the cAMP signaling assay. Therefore, the observed reduction in cAMP signaling can be attributed to the activity of the mutant receptors.

To address the reviewer's comment, we measured the basal activity of the I102A/I421A double mutant. The results showed that the double mutant exhibits basal activity comparable to that of the wild-type receptor (slightly higher), suggesting that the mutations do not abolish and increase the basal activity

of C3aR. As a control, we conducted the same assay with C5aR1 and its corresponding double mutant (I116A/M120A), as previously reported (Feng Y et al, 2023). The results confirmed that the assay conditions were appropriate for detecting basal activity.

Figure for referees not shown.

Construct	Mean	SEM	N
C3aR WT	100.0	0.0	5
C3aR I102A/I421A	98.0	3.3	4
C5aR1	109.2	7.1	3
C5aR1 I116A/M120A	78.4	2.8	5

Upon ligand binding, I102 interacts with the 4-chlorophenyl ring (Ring 2) of JR14a or with L75 of C3a as TM3 undergoes rotational and translational shifts. However, due to extensive interactions formed by other regions of the ligands, the I102A mutation did not significantly reduce the cAMP response induced by either ligand. In contrast, I421 interacts with A76 of C3a and establishes extensive interactions with both the 4-chlorophenyl ring and the 3-methylthiophene of JR14a. The I421A mutant exhibited a dramatic reduction in cAMP response induced by JR14a but not by C3a.

The I102A/I421A double mutant showed a significant decrease in cAMP response to JR14a, greater decrease than observed in I421A, due to the more loss of interactions with JR14a. Additionally, it exhibited a notable reduction in cAMP response to C3a. This is likely because the double mutation disrupts interactions not only with L75 but also with A76 of C3a, leading to a greater reduction than observed in either the I102A or I421A single mutants.

This difference in results can be attributed to variations in the binding network between C3a and JR14a. This point has been addressed on pages 10 and 15-16.

Page 10, lines 209-214,

“Notably, while I421^{7.39} forms van der Waals contact with A76 of C3a, it engages in more extensive interactions with the 4-chlorophenyl ring (Ring 1) and the 3-methylthiophene moiety of JR14a. In line with this, the I421^{7.39}A mutant exhibited a minor effect on C3a-mediated signaling but caused a 60-fold reduction in JR14a-induced signaling, underscoring the important role of I421^{7.39} in JR14a binding (Fig 2D, Fig EV5, Appendix Fig S4 and Appendix Table S1).

Pages 15-16, lines 344-349,

“The small-molecule agonist JR14a differs from C3a in size and interaction networks with C3aR.

While JR14a relies on specific interactions with residues within the orthosteric pocket, such as I421^{7,39}, C3a forms extensive contacts across both extracellular and transmembrane regions, likely stabilizing the active conformation through a broader network of interactions. These differences suggest that the structural contributions of individual residues to receptor activation vary depending on the ligand, making it difficult to define a single key residue responsible for C3aR activation.”

11. Minor comment: try replacing "constitutional activity" by "constitutive activity"

We thank the reviewer's comment.

Page 15, line 342, *“unlike C5aR1, C3aR exhibits **constitutive** activity, likely due to the lower stability”*

Referee #3:

This manuscript reported the cryo-EM structures of C3aR in complex with a small molecule synthetic agonist JR14a in two distinct conformational states: an intermediate state and a fully active Gi-bound state. Also, the authors presented the apo structure of C3aR and offered insights into its inactive conformation, which remain elusive till now. Comparison of the apo, intermediate, and fully active structures provide a comprehensive understanding of the activation mechanism of C3aR. This paper advances the field of structure-based drug design by providing valuable insights into the conformational landscape of C3aR. These information are significant to the field. The paper is organised well overall, and the results are presented with clarity and substantiated by strong analytical techniques. Therefore this manuscript is recommended for the publication in The EMBO Journal.

Comments:

"Recent structural studies of C3aR and C5aR1 have greatly advanced our understanding of anaphylatoxin recognition by their receptors. The active structures of C5a-bound C5aR1 coupled to Gi and the inactive structure of antagonist-bound C5aR1 have revealed specific ligand recognition and the conformational changes associated with C5aR1 activation¹⁰⁻¹³."

Comment: recent research of C5aR should be updated.

J Struct Biol. 2024 Sep;216(3):108117.

We thank the reviewer's comment. We have updated references.

Page 3, lines 45-48,

"The active structures of C5a-bound C5aR1 coupled to Gi and the inactive structure of antagonist-bound C5aR1 have revealed specific ligand recognition and the conformational changes associated with C5aR1 activation (Feng et al, 2023; Wang et al, 2023; Yadav et al, 2023; Liu et al, 2018; Yang et al, 2024)."

"Our findings provide valuable insights for structure-based drug design targeting C3aR..."

The authors should discuss more about how the data presented could be informative to develop C3aR-targeting drugs.

We thank the reviewer for the comment. We have included the following statement in the Discussion section.

Page 16, lines 357-365,

"Our findings provide valuable insights for structure-based drug design targeting C3aR. The binding site and the specific interactions identified in the JR14a-bound structures offer crucial information for developing more potent and selective small-molecule C3aR agonists or antagonists. Furthermore, our apo structure highlights the inactive conformation of C3aR, offering a foundation for designing antagonists that stabilize this inactive conformation. The intermediate conformation captured in the JR14a-C3aR-BRIL structure demonstrates early conformational changes induced by agonist binding and provides insights into the transitions required for receptor activation. By integrating insights from the apo, intermediate, and fully active states of C3aR, these structural data establish a framework for optimizing agonists or antagonists to achieve the desired therapeutic outcomes."

Dear Hee-Jung,

Thank you again for submitting your revised manuscript EMBOJ-2024-119203R to The EMBO Journal for our consideration. It has been sent back to two of the original referees that had previously assessed the first version of your manuscript, and we have now received their comments, which are included below.

I am very pleased to say that both referees recognize that the initially raised concerns have been fully addressed in the thoroughly revised manuscript, and they now support its publication. There are only a few minor errors pointed out by referee #1, which we kindly ask you to correct in a final version of your manuscript.

I would also like to remind you that the recently published work reporting the structure of the JR14a-C3aR-Gi complex should be cited and also discussed in your manuscript where appropriate, as we have previously discussed.

From the editorial side, there are also a few corrections and changes we need you to make in the final version before we can formally accept it for publication in The EMBO Journal. Please include in your resubmission a brief cover letter describing how the points below are addressed:

- Thank you for providing the referees with confidential access to the PDB files, validation reports, and map files during peer review. The following temporary statement can now be removed from the Data availability section: "For the information of the referees during peer review of the manuscript, the PDB files along with their validation reports have been uploaded to the journal's manuscript tracking system where they can confidentially be accessed from. Due to their large sizes, the map files can be accessed from the external link: link". Please make sure that the permanent links (URLs) to all deposited datasets are included in the Data availability statement, and that all datasets (9IPY, 9IPV, EMD 60836, EMD 60785, EMD 60782, EMD 60783, EMD 60784) will be publicly available at the time of publication.

- Please provide a list of up to 5 relevant keywords after the Abstract of your revised manuscript.

- Please change the heading "Competing Interests statement" to "Disclosure and competing interests statement".

- The author contributions statement should be removed from the manuscript file. Instead, we use CRediT to specify the contributions of each author in the journal submission system. Please feel free to use the free text box to provide more detailed descriptions during submission. See also our guide to authors for more information: <https://www.embopress.org/page/journal/14602075/authorguide#authorshipguidelines>.

- The callouts for Appendix Table 1 and Supplementary Table 1 should be corrected to "Appendix Table S1".

- We noticed that callouts are missing for Appendix Figure S7; please make sure that all Figures and their panels are called out in the revised manuscript.

- The References in the Appendix PDF file should be listed alphabetically.

- Please reorganize your Source Data files in one zip folder per main Figure (e.g., all Source Data files for Figure 1 panels should be saved in a single folder, which then needs to be zipped and uploaded as "SD Figure 1.zip"). For Expanded View and Appendix Figures, please zip all their Source Data together in a single "SD EV Appendix Figures.zip" folder.

- The dimensions of the synopsis image (in .jpg or .png) format should be exactly 550 pixels wide and 300-600 pixels high (the height is variable). Please make sure that all text will be legible at this final size (some fonts in the current version of the image are too small).

- Please note that EMBO press papers are also accompanied online by:

A) a short (2 sentences) summary of the findings and their significance, and

B) 2-5 short bullet points highlighting the key results.

Please upload this information in a separate Word file.

- During our routine pre-acceptance checks, our data editors have raised the following queries regarding figures, data, and legends. Please make sure that all requests below are completely addressed in the final version of your manuscript:

1) Please provide the exact p values in the legends of Figures 2D.

2) Please note that in Figure 2D there is a mismatch between the annotated p values in the figure legend and the annotated p values in the figure file that should be corrected.

- The manuscript section order should be corrected as follows: Title page - Abstract & Keywords - Introduction - Results - Discussion - Methods - Data Availability - Acknowledgements - Disclosure and Competing Interests Statement - References -

Figure Legends - main Table(s) (if there are any) - Expanded View Figure Legends.

Please also note that as part of the EMBO publications' Transparent Editorial Process, The EMBO Journal publishes online a Peer Review File along with each accepted manuscript. This File will be published in conjunction with your paper and will include the referee reports, your point-by-point response and all pertinent correspondence relating to the manuscript. You can opt out of this by letting the editorial office know (contact@embojournal.org). If you do opt out, the Peer Review File link will point to the following statement: "No Peer Review File is available with this article, as the authors have chosen not to make the review process public in this case."

We look forward to seeing a final version of your manuscript as soon as possible. Please let us know if you have any questions and use this link to submit your revision: <https://emboj.msubmit.net/cgi-bin/main.plex>.

Best regards,

Ioannis

Referee #1:

The authors have incorporated the suggestions in the extensively revised manuscript and it reads very well. I noticed a few typographical errors that need correcting.

Line 99, 'data' to 'date'.
References are not listed in alphabetical order
Line 788, delete 'For model'

I do not need to referee these corrections.

Referee #2:

The authors have meticulously addressed the issues. The revised manuscript is considerably improved with added data and explanation. Therefore, I recommend that the editor may take a favorable decision.

All editorial and formatting issues were resolved by the authors.

Referee #1:

The authors have incorporated the suggestions in the extensively revised manuscript and it reads very well. I noticed a few typographical errors that need correcting.

We appreciate the reviewer's careful review of our revised manuscript. The errors they identified have been corrected.

Line 99, 'data' to 'date'.

→ Line 100; "data" has been corrected to "date"

References are not listed in alphabetical order

→ The references have been listed alphabetically

Line 788, delete 'For model'

→ Line 765; 'For model' has been deleted.

Referee #2:

The authors have meticulously addressed the issues. The revised manuscript is considerably improved with added data and explanation. Therefore, I recommend that the editor may take a favorable decision.

We appreciate the reviewer's comment.

Other changes

As one of our authors (Jinuk Kim) recently changed his affiliation, we have updated his information accordingly.

Lines 10-11: His current address has been added.

Dear Hee-Jung,

Congratulations on an excellent work! I am very pleased to inform you that your manuscript has now been accepted for publication in The EMBO Journal. Thank you for your comprehensive responses to the referees' comments and concerns, and for addressing our editorial and formatting requests.

If you have any questions, please do not hesitate to contact the Editorial Office. Thank you for your contribution to The EMBO Journal. Working with you has been a pleasure!

Best regards,

Ioannis
